# Direct observation of 3D nitrogen distribution in silicon-based dielectrics using atom probe tomography

Byeong-Gyu Chae [1] ✉, Jeong Yeon Won[1], Young Sik Shin[1], Dong Jin Yun [1], Jae min Ahn[1], Seon Tae Park[1], Ki-bum Lee[2], Hokyun An[2], Mina Seol[2], I-Jun Ro[3], Se-Ho Kim[3], Chunhyung Chung[2] & Eunha Lee [1] ✉

The distribution of nitrogen in semiconductor devices plays a crucial role in tuning their physical and electrical properties. However, direct observation and precise quantification of nitrogen remain challenging because of analytical limitations, particularly at critical interfaces in silicon-based semiconductors. Although atom probe tomography has emerged as a powerful tool, distinguishing nitrogen from silicon without isotope doping is persistently difficult. In this study, we employ advanced atom probe tomography with an extended flight path under optimized conditions to characterize the three-dimensional nitrogen distribution in actual device structures, including 2- and 5-nm-thick silicon dioxide/silicon oxynitride-based gate dielectrics and a fin-structured three-dimensional device. Our analysis reveals that the nitrogen distribution determines the formation of the nitrogen profile in gate dielectrics, which in turn affects the diffusion of impurities, ultimately impacting the electrical properties and reliability. Our work provides insights into atomic-scale nitrogen behavior, paving the way for advancing next-generation semiconductor devices.

N has been widely incorporated into high-performance semiconductor devices as its distribution and concentration can reliably modulate their physical and electrical properties. As a result, extensive research has focused on manipulating the N behavior in Si-based semiconductors, including its effects on band gap engineering[1–4], trap density control[5–8], and passivation enhancement[9–11]. Understanding these effects requires directly observing the N distribution; however, conventional analytical techniques face significant limitations in providing accurate three-dimensional (3D) quantification.

Although transmission electron microscopy (TEM) has been used to investigate the behavior of N, especially in gate dielectrics formed by plasma nitridation (PN), it has limited capability in observing and quantifying light elements, such as N and O, especially at low concentrations[12–17]. Secondary-ion mass spectrometry (SIMS) is an important reference tool for estimating N profiles; however, although SIMS provides high sensitivity for low-level detection, it only provides a 1D depth profile[18–20]. These constraints hinder a comprehensive understanding of how N influences device performance, emphasizing the need for advanced analytical methods. In this study, we employ atom probe tomography (APT), which provides quantitative 3D compositional information with atomic-scale spatial resolution[21–23]. APT has been extensively applied in diverse areas, including semiconductor and optoelectronic devices, dielectric materials, and nanowires[24–34]. However, the overlaps between the Si and N mass peaks in APT mass spectra have typically made it difficult to accurately characterize the behavior of N in Si-based semiconductors[35–39]. Some studies have tracked N behavior using isotopically labeled dopants such as $^{15}$N or $^{30}$Si, but this approach does not accurately represent its behavior in

[1]Analytical Engineering Group, Material Research Center, Samsung Advanced Institute of Technology, Samsung Electronics Co., Ltd, Suwon, Republic of Korea. [2]Process Development Team, Semiconductor R&D Center, Samsung Electronics Co., Ltd, Hwaseong, Republic of Korea. [3]Department of Materials Science & Engineering, Korea University, Seoul, Republic of Korea. ✉e-mail: bg.chae@samsung.com; eunhayo.lee@samsung.com

actual device structures[36,40]. To overcome this issue, we utilize the advanced Invizo 6000 atom probe, which features a multiple Einzel lens system and a longer flight path length than prior instruments. This work demonstrates APT-based analysis of N in Si-based device structures without isotope doping, which is not commonly reported.

In this study, we directly observe and quantify the N distribution to determine the influence of the N distribution on the properties of $SiO_2/SiON$ gate dielectrics using APT. $SiO_2/SiON/TiN$ and $SiO_2/SiON/B$-doped poly-Si structures with thin and ultra-thin gate dielectrics of 5 nm and 2 nm, respectively, are prepared for applications in two different semiconductor devices, namely, dynamic random access memory (DRAM) and logic devices. Specifically, the distribution of N incorporated by PN is quantified using APT. In the thin gate dielectric, the formation of a SiON layer with a higher N concentration improves the performance of the material in preventing impurities from reaching the gate electrode. Similarly, the electrical properties of the ultra-thin gate dielectric are improved by the formation of a SiON layer, which exhibits a shallower N profile but higher N concentrations. This property is strongly correlated with the PN technique used to modify the N profile. The penetration of the B dopant from the gate electrode is also significantly suppressed by changing the N profile. Our findings enable the quantification of the 3D N distribution and thus suggest pathways for the development of next-generation semiconductor technologies.

## Results and discussion
### Direct observation of nitrogen in silicon-based materials
The feasibility of directly characterizing N in semiconductor materials using APT is assessed using a 30-nm-thick SiN film, which is widely used in semiconductors owing to its superior insulating properties[41–43]. SiN has a wide range of applications, from gate dielectrics in DRAM[8,44,45] to logic devices and floating gates in NAND flash memory devices[46–59]. Conventional APT, such as the local electrode atom probe (LEAP), has difficulty distinguishing N from Si in Si-based materials because of the peak overlap in the mass spectrum, especially in the absence of isotope doping. LEAP has a rather low mass-resolving power, resulting in mass spectra with overlapping peaks corresponding to different elements with a similar mass-to-charge ratio. To distinguish N from Si without isotope doping, a mass resolution of approximately $m/\Delta m \approx 1400$ is required in APT, considering the peak broadening effects inherent to the technique (Supplementary Table 1). Detecting N in Si-based materials is therefore particularly challenging, despite its necessity in semiconductor applications; however, under optimized analytical conditions, the Invizo 6000 can differentiate between Si and N ions owing to its improved mass-resolving power (Fig. 1, Supplementary Table 1 and Supplementary Fig. 1). Figure 1a shows a 3D atom map of a 30-nm-thick SiN layer deposited on a Si

substrate. Although the mass peaks of $Si^+$ and $N_2^+$ (shown in gray and green, respectively) partially overlap, they are sufficiently separated to identify each ion (Fig. 1b and Supplementary Figs. 1 and 2). The 3D elemental maps of Si and N on the 30-nm-thick SiN are shown in Fig. 1c and 1d, respectively. N is observed as both $N^+$ and $N_2^+$; however, for visualization, only $N_2^+$ is used as its peak better separates from that of Si in the APT mass spectrum. Despite having approximately the same mass-to-charge ratio, the distributions of $N_2^+$ and $Si^+$ are not correlated. As expected, $N_2^+$ is located in the SiN layer, while $Si^+$ is distributed at a relatively low density in the SiN layer and native $SiO_2$ layer but at a high density on the Si substrate. The variation in the mass peak intensities between $Si^+$ and $N_2^+$ across a laser pulse energy range of 25–100 pJ further corroborates the presence of two distinct elements (Supplementary Fig. 3). Consequently, our results demonstrate that APT can directly characterize the N distribution in Si-based materials.

### Nitrogen distribution in thin gate dielectrics
To enhance the electrical performance concurrently with the gate length reduction in semiconductor devices, the gate dielectric needs to be reduced; however, the gate leakage current increases exponentially as the gate dielectric becomes thinner[50–52]. The $SiO_2$ gate dielectric is commonly replaced with SiON or nitrided $SiO_2$ to overcome this problem because the gate leakage current can be reduced by incorporating N[53–57]. Optimizing the distribution of N in the gate dielectric is therefore important to reduce the gate leakage current and suppress other detrimental effects. In particular, PN technology is key for effectively controlling the N distribution, followed by the formation of a $SiO_2/SiON$ gate dielectric[58–63]. Despite the many studies on $SiO_2/SiON$ gate dielectrics, the effects of N distribution on the properties of $SiO_2/SiON$ gate dielectrics are poorly understood owing to the lack of viable analysis techniques. To address this issue, we investigated the N distribution in the $SiO_2/SiON/TiN$ structure for DRAM device applications under different PN operating parameters.

Three different samples were prepared to study the effects of PN and its processing parameters on $SiO_2/SiON$ gate dielectrics. To that end, one gate dielectric was prepared without PN, and two others were prepared using different PN processes. Specifically, for one sample, PN was conducted for a short PN time ($t_{PN}$) followed by post-nitridation annealing (PNA) for a time ($t_{PNA}$) under $O_2$, and for the other, PN was performed for a longer time ($1.5 t_{PN}$) followed by PNA for $t_{PNA}$ under $N_2$. Different SiON layers were formed on the $SiO_2$ gate dielectric by employing an appropriate PN time, PNA time, and ambient atmosphere, as outlined in numerous studies[60–63]. High-resolution scanning transmission electron microscopy (HR-STEM) images show $SiO_2/SiON$ gate dielectrics with thicknesses ranging from ~5.0 to 5.3 nm (Fig. 2). A

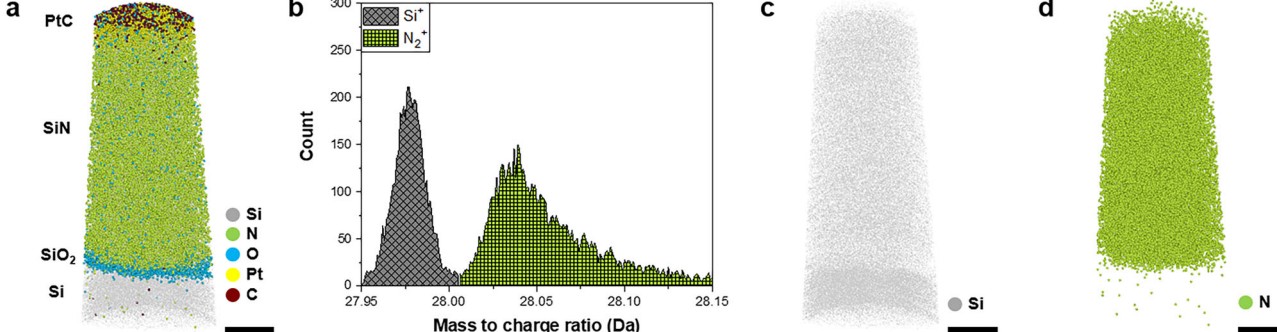

**Fig. 1 | APT analysis of 25-nm-thick SiN film. a** 3D reconstruction of Si/$SiO_2$/SiN stack. **b** Invizo 6000 mass spectra of $Si^+$(gray) and $N_2^+$(green). The $Si^+$ peak is distinguished from the $N_2^+$ peak in the APT mass spectrum. 3D ion map of **c** Si and **d** N demonstrating the direct characterization of the N behavior in Si-based materials using the Invizo 6000. (Scale bars: 5 nm in **a**, **c**, **d**).

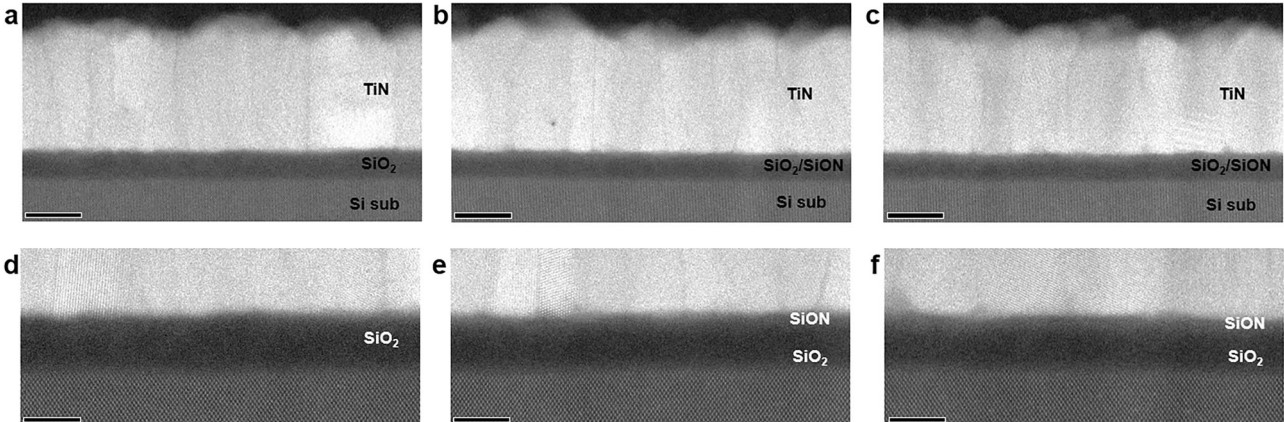

**Fig. 2 | HAADF-STEM cross-sectional images of SiO$_2$/SiON/TiN structure.** **a–c** Low- and **d–f** high-magnification HAADF-STEM images of structures obtained with various PN and PNA processes: **a**, **d** without PN, **b**, **e** after PN for $t_{PN}$ followed by PNA for $t_{PNA}$, and **c**, **f** after PN for $1.5t_{PN}$ followed by PNA for $t_{PNA}$. A contrast between the relatively bright SiON layer and the duller SiO$_2$ layer is observed after PN. A 23-nm-thick TiN layer is deposited on top of the gate dielectric as a gate electrode. The precise N content and its distribution are not visible (Scale bars: 10 nm in **a–c**, 5 nm in **d–f**).

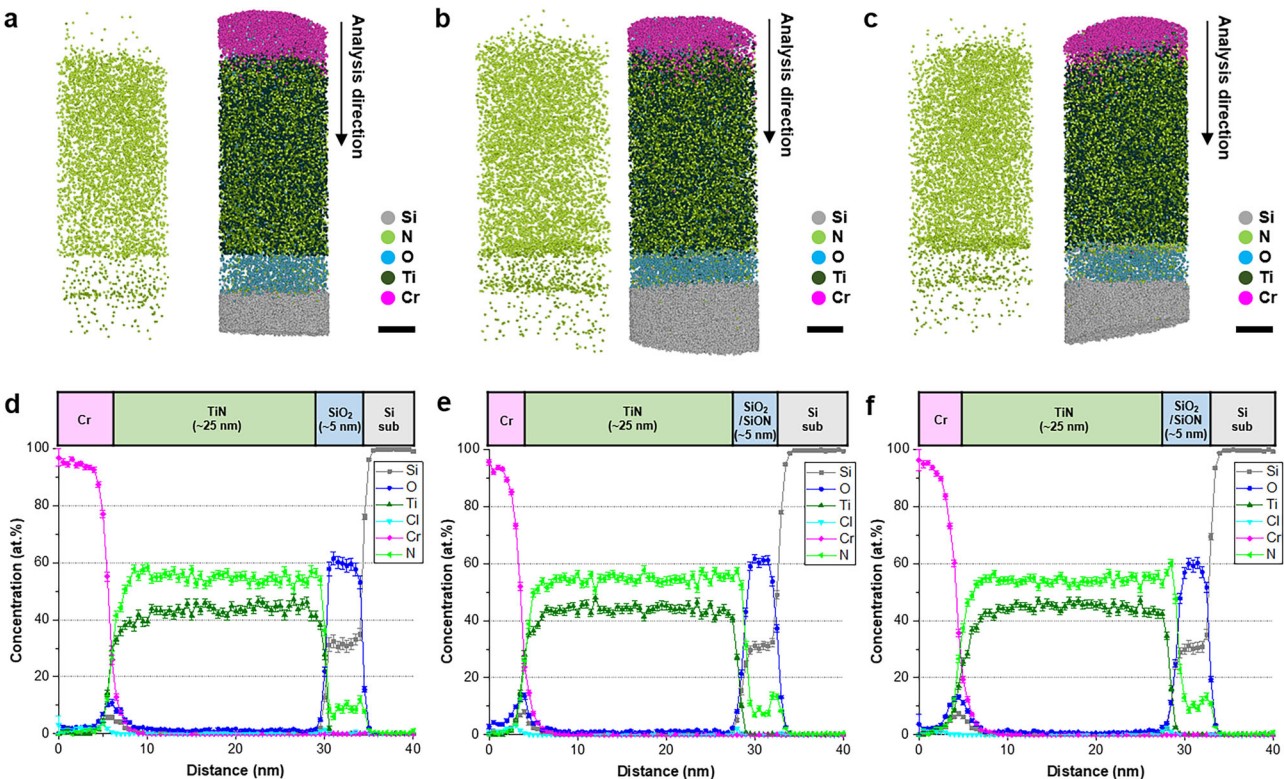

**Fig. 3 | APT analysis of the Si/SiO$_2$/SiON/TiN structure. a–c** 3D ion maps and **d–f** 2D composition profile of the sample **a**, **d** without PN, **b**, **e** after PN for $t_{PN}$ followed by PNA for $t_{PNA}$ (**c**, **f**), and after PN for $1.5t_{PN}$, followed by PNA for $t_{PNA}$. The N incorporated by PN is clearly visible at the SiO$_2$/TiN interface in both the 3D ion maps and the 2D composition profiles. The N concentration at the interface increases with the PN time. The accumulation of N at the Si sub/SiO$_2$ interface is attributed to N diffusion from the TiN gate electrode during annealing (Scale bars: 5 nm in **a–c**). Error bars in (**d–f**) indicate 1-sigma counting statistics.

gate electrode consisting of a 25-nm-thick TiN layer was deposited on top of the SiO$_2$/SiON gate dielectric (Fig. 2a–c). In the magnified high-angle annular dark-field STEM (HAADF-STEM) images, in which the intensity is approximately proportional to the square of the atomic number, the SiON layer appears somewhat lighter than the SiO$_2$ layer (Fig. 2d–f); however, apart from this difference in contrast, obtaining information regarding the amount and 3D distribution of N proved difficult.

Accordingly, the 3D distribution of N and the resulting distribution of impurities in the SiO$_2$/SiON/TiN structure were characterized using APT. The reconstructed 3D atom maps of the SiO$_2$/SiON/TiN structure under different PN operating parameters reveal the formation of a SiON layer on the SiO$_2$ layer after PN (Fig. 3a–c and Supplementary Fig. 4). N, which is detected as both N$^+$ and N$_2$$^+$ in APT, was primarily located between the SiO$_2$ gate dielectric and the TiN metal gate in the upper part of the 5-nm-thick gate dielectric (Fig. 3a–c). The

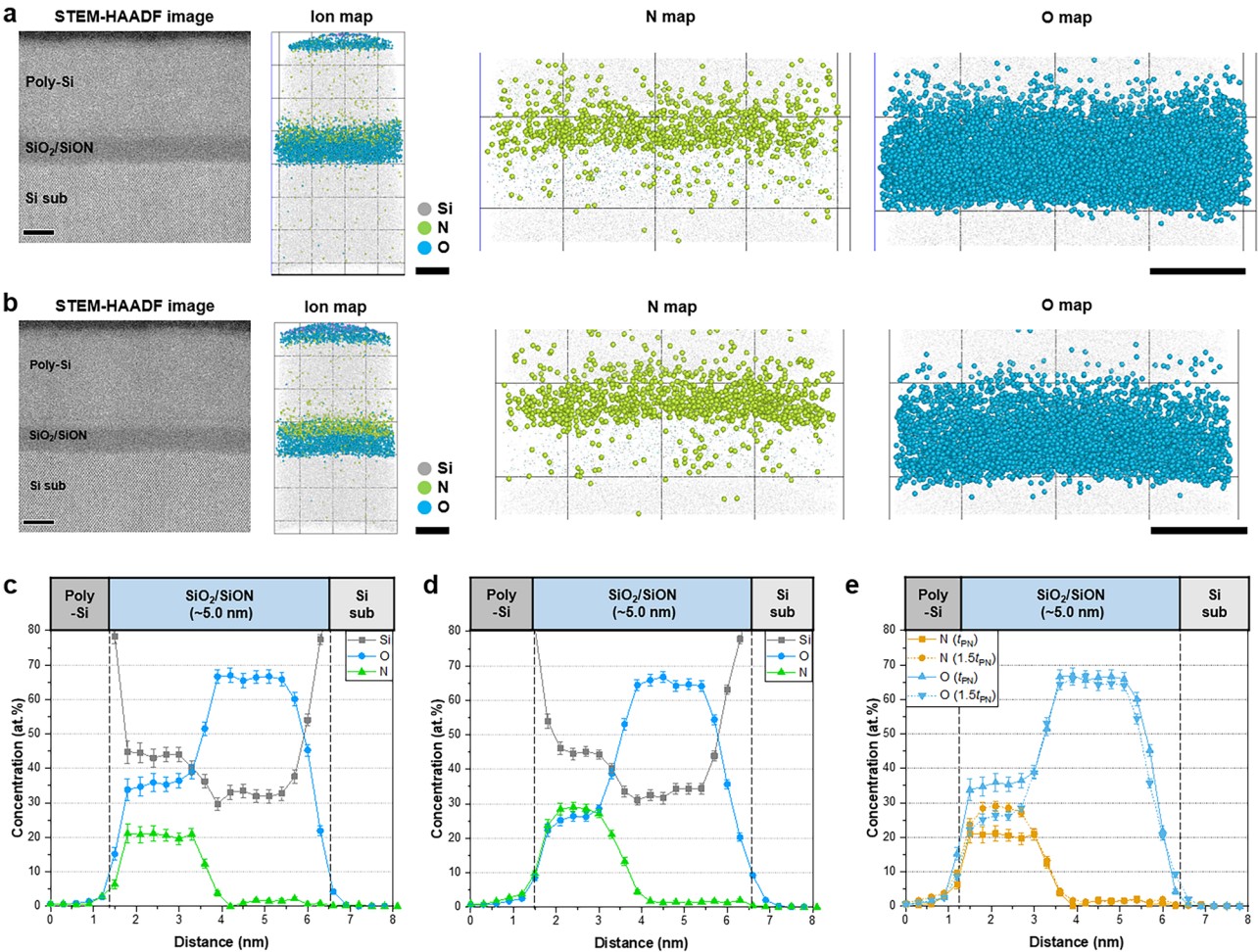

**Fig. 4 | APT analysis of Si/SiO₂/SiON/poly-Si structure.** HAADF-STEM cross-sectional image and APT 3D ion maps of the sample **a** after PN for $t_{PN}$ followed by PNA for $t_{PNA}$ under $O_2$ and **b** after PN for $1.5t_{PN}$ followed by PNA for $t_{PNA}$ under $N_2$. The N incorporated by PN is clearly visible. **c**, **d** 2D composition and **e** N and O profiles measured under different PN and PNA conditions. The robust SiON layer formed at a longer $t_{PN}$ is clearly visible (Scale bars: 5 nm in **a**, **b**). Error bars in (**c**–**e**) indicate 1-sigma counting statistics.

3D N map clearly reveals a higher N concentration near the $SiO_2$/TiN interface after PN, not only providing the concentration but also a 3D ion map that facilitates a more in-depth investigation of the influence of the precise location of N and impurities on the material properties. Before PN, the N concentration at the $SiO_2$/TiN interface decreased abruptly along with the Ti concentration (Fig. 3d–f and Supplementary Fig. 5). Conversely, after PN, the N profile at the dielectric/TiN interface was broader than that of Ti, indicating that a sufficient concentration of N was located in the upper part of the gate dielectric to form a SiON layer. The N concentration at the dielectric/TiN interface also increased with the PN time. The maximum N concentration in the SiON layer reached ~58 at.% after a short PN time ($t_{PN}$) and increased to more than 60 at.% after prolonged PN ($1.5t_{PN}$). The highest concentration of N was incorporated in the SiON layer during PN, and this concentration gradually decreased toward the $SiO_2$ layer. The change in the N concentration as a function of the PN time shows good agreement with the X-ray photoelectron spectroscopy, SIMS, and STEM-electron energy loss spectroscopy (STEM-EELS) results (Supplementary Figs. 6–8). The 3D N map and composition profile show that a significant amount of N accumulates at the $Si/SiO_2$ interface, even in the structure without PN (Fig. 3). The tendency of N to accumulate at the $Si/SiO_2$ interface was attributed to the TiN annealing process rather than the PN process. Without the TiN annealing process, only a small amount of N would reach this region through thermal diffusion during PNA. These results

demonstrate the ability of APT to clearly distinguish the SiON and $SiO_2$ layers by directly observing the N distribution. This paper demonstrates the visualization and quantification of N in Si-based semiconductors without isotope doping, which remains challenging.

We also investigate the 3D N distribution in the gate dielectric/poly-Si stack structure, as its interface is one of the most important among the numerous interfaces in DRAM devices. Notably, in the APT mass spectrum of the $SiO_2$/SiON/poly-Si structure, only the $N_2^+$ peak was differentiated from that of $Si^+$, whereas the $N^+$ peak was further differentiated from that of $Si^{2+}$ for the $SiO_2$/SiON/TiN structure owing to the significant amount of N originating from the TiN gate and the SiON layer in this structure. Upon replacing the TiN gate with poly-Si, the $N^+$ peak is obscured by overlap with the $Si^{2+}$ peak; thus, the $N^+$ peak is excluded, and the precise distribution of N in the $SiO_2$/SiON/poly-Si structure was determined solely from the $N_2^+$ peak. Although this method underestimated the N content, the formation of a SiON layer on the $SiO_2$ layer was more clearly demonstrated (Fig. 4a, b). Like that of the $SiO_2$/SiON/TiN structure, the N concentration in the SiON layer changed significantly with the increasing PN time (Fig. 4c, d). The SIMS and STEM-EELS results correspond well to the APT data, further supporting the reliability of N detection (Supplementary Figs. 9 and 10). The SiON layer contained ~20 at.% N after a short PN time but more than 28 at.% after a longer PN time under similar charge state ratio (CSR) values (Supplementary Fig. 11). Because APT field evaporation

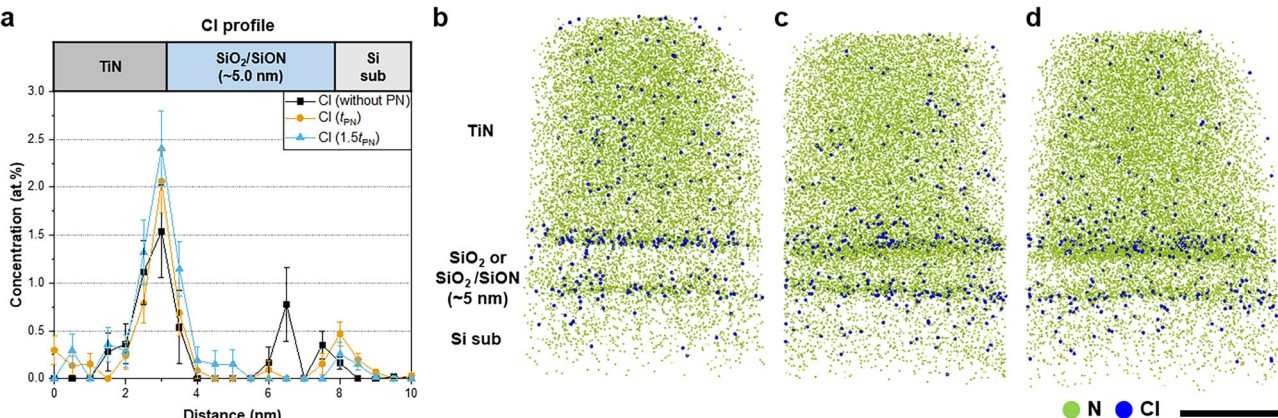

**Fig. 5 | APT analysis of residual Cl. a** Cl profile and **b–d** Cl map of the samples **b** without PN, **c** after PN for $t_{PN}$ followed by PNA for $t_{PNA}$, **d** and after PN for $1.5t_{PN}$ followed by PNA for $t_{PNA}$. Residual Cl originating from the TiCl$_4$ source accumulates at the interface. PN reduced the amount of residual Cl inside the gate dielectrics. (Scale bars: 10 nm in **b–d**). Error bars in (**a**) indicate 1-sigma counting statistics.

conditions can vary depending on the analysis environment, comparing data at similar CSR values—an indicator of comparable field conditions—is preferable for ensuring a reliable compositional analysis[18,64–66]. We expect that a longer PN time would incorporate more N into the gate dielectric, as N primarily interacts with SiO$_2$ during PN through O substitution or interstitial diffusion. However, because of the strong Si–O bonding, N diffusion into the deeper SiO$_2$ region is significantly restricted, as shown in the N profile. As a result, the N concentration remains highest near the upper part of the SiO$_2$ layer, creating a gradual N profile with a diffusion tail extending into the SiO$_2$. The thermal budget provided by the PNA process enhances atomic mobility, promoting N diffusion and enabling N to penetrate slightly deeper into the SiO$_2$. The O concentration in the SiON layer decreased as more N was incorporated, but remained constant in the SiO$_2$ layer (Fig. 4e). Our APT results directly confirm the formation of the SiON layer in the SiO$_2$/SiON/poly-Si structure and the difference in the N distribution as a function of the PN time.

We further investigate the effect of the N distribution formed by PN on the distribution and content of impurities in the SiO$_2$/SiON/TiN structure. A significant amount of Cl is present in the TiN. Before PN, this Cl accumulates at the SiO$_2$/TiN interface, diffuses into the SiO$_2$ layer, and spreads throughout the SiO$_2$ layer (Fig. 5a). In contrast, after PN, Cl is highly concentrated at the SiON/TiN interface but was almost entirely absent in the SiO$_2$ layer (Fig. 5b–d). Cl is known to have a detrimental effect on semiconductor stacks because it acts as an interface trap at the Si/SiO$_2$ interface or an oxide trap in the SiO$_2$ insulator[64,65]. Cl primarily originates from the TiCl$_4$ used in TiN deposition and diffuses toward the gate dielectric during TiN annealing. The formation of a SiON layer on SiO$_2$ after PN prevents Cl from penetrating the gate dielectric (Fig. 5). The Cl concentration at the dielectric/TiN interface increases from ~1.5 at.% before PN to more than 2.0 at.% after PN; thus, the prolonged application of PN promotes the accumulation of Cl at the dielectric/TiN interface and reduced the amount of Cl inside the gate dielectric, thereby enhancing its effectiveness as a diffusion barrier. A small amount of Cl was also observed at the Si/SiO$_2$ interface after PN. Engineering the N profile contributes to reducing the number of Cl-induced trap sites at the Si/SiO$_2$ interface and SiO$_2$/SiON insulator. Cl that penetrates the gate dielectric can either form a dangling bond that attracts electrons at the interface or act as a mobile ion in the insulator, thereby inhibiting charge transfer. These impurity-induced defects can degrade device performance by disrupting charge transport, increasing leakage currents, and accelerating failure mechanisms such as time-dependent dielectric breakdown (TDDB) and negative bias temperature instability (NBTI)[67,68].

Accordingly, preventing the penetration of impurities is essential to improve their properties. The trap density and the resulting degradation in reliability can be effectively suppressed through characterization and modification of the N distribution.

## Nitrogen distribution in ultra-thin gate dielectrics
Although incorporating N reduces the leakage current into the gate dielectric, it also degrades the NBTI performance in ultra-thin SiO$_2$/SiON gate dielectrics[69–71]. Therefore, considerable effort has been devoted to improving the NBTI performance by adjusting the PN process to produce the desired N distribution. However, success has proven elusive owing to the considerable challenge of estimating the N distribution in 3D semiconductor devices; this has hindered a deeper understanding of the behavior of N in ultra-thin gate dielectrics, which is crucial for the development of logic devices and next-generation DRAM devices.

To address this issue, we use the aforementioned analytical technique to compare the quantitative 3D N distribution in ultra-thin SiO$_2$/SiON gate dielectrics produced by the two different PN processes: PN for $t_{PN}$, followed by PNA for $t_{PNA}$ under O$_2$, and PN for $1.5t_{PN}$, then PNA for $0.15t_{PNA}$ under N$_2$. HR-STEM images confirm the formation of SiON and SiO$_2$ layers with thicknesses ranging from 1.1 to 1.2 nm (Fig. 6a, b). A gate electrode consisting of a layer of B-doped poly-Si measuring ~8.5 nm thick was deposited on the SiO$_2$/SiON structure. Although the SiO$_2$ and SiON layers were distinguishable, observing the 3D N distribution and quantifying the concentration prove difficult. In particular, a comparison of the N distribution was more challenging because the ultra-thin SiO$_2$/SiON gate dielectrics were highly vulnerable to electron beams during STEM.

The effects of the N distribution on the electrical properties of the ultra-thin SiO$_2$/SiON gate dielectrics are explored using APT. The N incorporated by PN created a SiON layer on the SiO$_2$ layer in the upper part of the gate dielectric (Fig. 6). Analysis of the same sample using LEAP fails to show the N distribution (Supplementary Fig. 12). The N content gradually decreased toward the lower part of the gate dielectric, which consisted of SiO$_2$. The 3D spatial distributions of the Si and N atoms in the ultra-thin gate dielectrics formed with different PN parameters are shown in Fig. 6a, b. The N$^+$ peak was obscured by overlap with the Si$^{2+}$ peak in the mass spectra of these structures; thus, their N distributions were evaluated using only the N$_2^+$ peak. The quantitative composition profiles along the depth show that the PN process had a considerable influence on the N distribution. The N concentration was the highest in the SiON layer, particularly after prolonged PN (Fig. 6c, d). With the short PN time, the N concentration

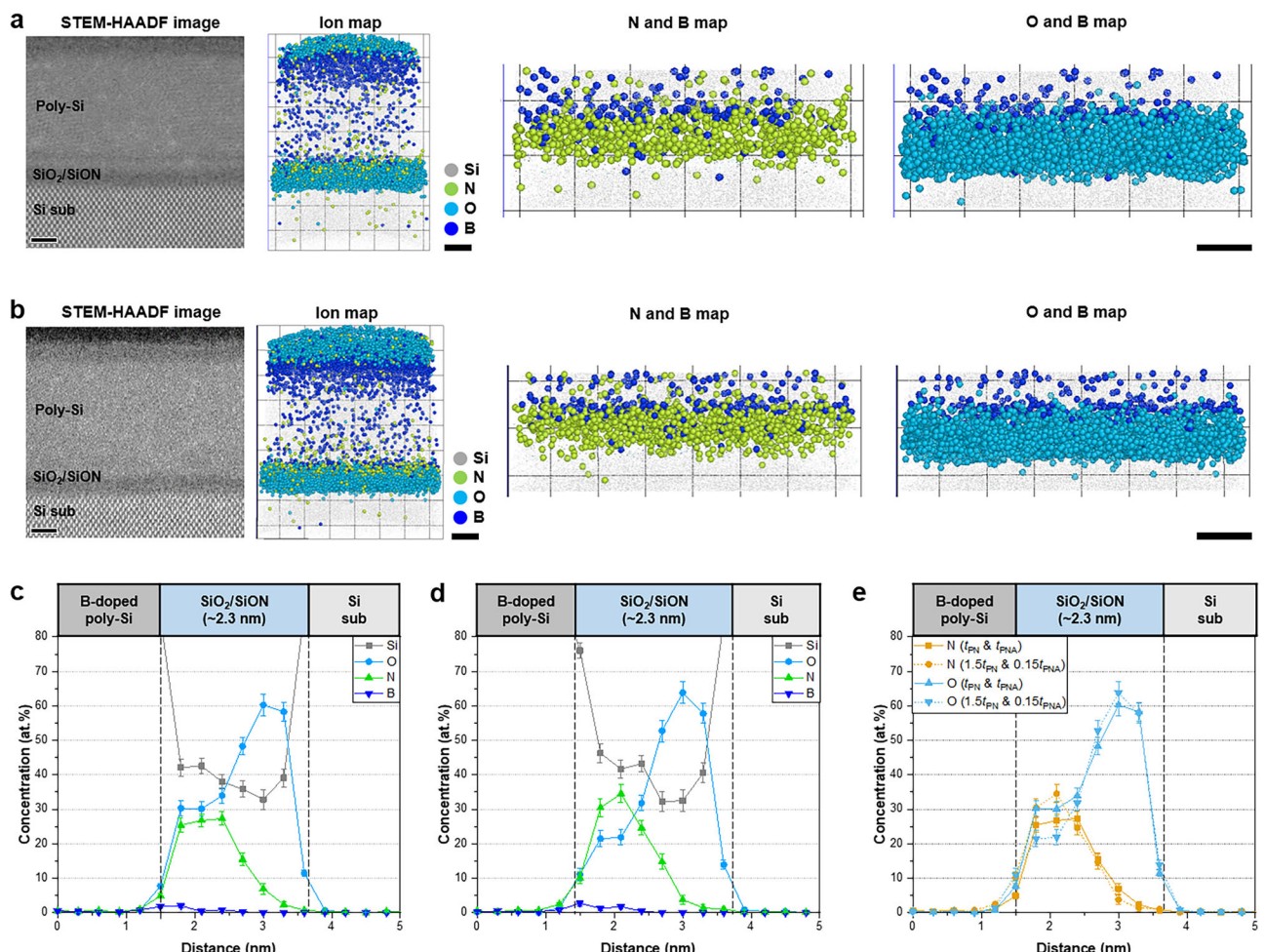

**Fig. 6 | APT analysis of the Si/SiO₂/SiON/B-doped poly-Si structure.** HAADF-STEM cross-sectional image and APT 3D ion maps of the sample **a** after PN for $t_{PN}$ followed by PNA for $t_{PNA}$ under $O_2$ and **b** after PN for $1.5t_{PN}$ followed by PNA for $0.15t_{PNA}$ under $N_2$. The SiON layer is clearly visible after PN, even in the ultra-thin gate dielectrics. **c**, **d** 2D composition profile and **e** N and O profiles measured under different PN and PNA conditions. A longer $t_{PN}$ and shorter $t_{PNA}$ under $N_2$ results in a shallower N profile and a higher N concentration in the SiON layer (Scale bars: 2 nm in **a**, **b**). Error bars in (**c**–**e**) indicate 1-sigma counting statistics.

in the SiON layer reached a maximum of 27 at.%, which increased to more than 30 at.% after a longer PN time. The difference of ~7.0 at.% was attributed to variations in the N content injected during the PN process; however, the lengths of the N diffusion pathways in the two samples differed significantly. Shorter PN and longer PNA times resulted in a broader N profile, indicating the diffusion of more N atoms to the interior (Fig. 6e). The APT N profiles showed that the N distribution was shallower at shorter PNA times, regardless of the PN time. Increasing the PN time increased the N concentration in the SiON layer, and the subsequent shorter PNA time suppressed N diffusion owing to the reduced heat budget. Hence, we confirm that simultaneously adjusting the PN and PNA processes resulted in a shallow N profile with a higher N concentration in the SiON layer. These APT results agree with the SIMS results, showing a stark difference in the N distribution with variations in the PN or PNA processes (Supplementary Fig. 13).

Engineering the N profile is key to improving the electrical properties of SiO₂/SiON/B-doped poly-Si structure. A longer PN time with a shorter PNA time significantly improved the NBTI performance of the structure, which exhibited an NBTI ~28.6% lower than that of the other structure. In contrast, the TDDB, which measures oxide breakdown during long-term applications, remained essentially unchanged. This result indicates that changing the N distribution can significantly improve the NBTI characteristics without the loss of TDDB, which is

one of the main aging-induced failure mechanisms. The formation of a fixed positive charge at the Si/SiO₂ interface is known to have a significant impact on the NBTI. The incorporation of N deteriorates the NBTI characteristics by generating fixed positive charges such as Si–NH⁺ at the Si/SiO₂ interface[72,73]. We found that minimizing N diffusion into the Si/SiO₂ interface suppressed the formation of fixed positive charges while maintaining a higher N concentration in the SiON layer.

The effect of the N profile on the blocking performance of the structure was also investigated. Prolonged PN times resulted in the accumulation of more B at the SiON/poly-Si interface (Fig. 6a, b), with B contents of ~2.0 and 2.7 at.% being observed after short and long PN times, respectively (Supplementary Fig. 14). In the interior of the gate dielectric, only a low concentration of B dopant diffused through the SiON layer after a longer PN time owing to the formation of a robust SiN diffusion barrier, which effectively suppressed B penetration into the gate dielectric. The APT-quantified results are consistent with the SIMS result (Supplementary Fig. 15). Although B is intentionally doped into the poly-Si gate to modulate the work function and optimize the electrical properties, it can degrade device reliability by increasing the gate leakage current if it diffuses into the gate dielectric. Our analytical approach confirms the importance of a robust SiON layer with a high N concentration to inhibit the diffusion of dopants and impurities. Because diffusion from the gate often has

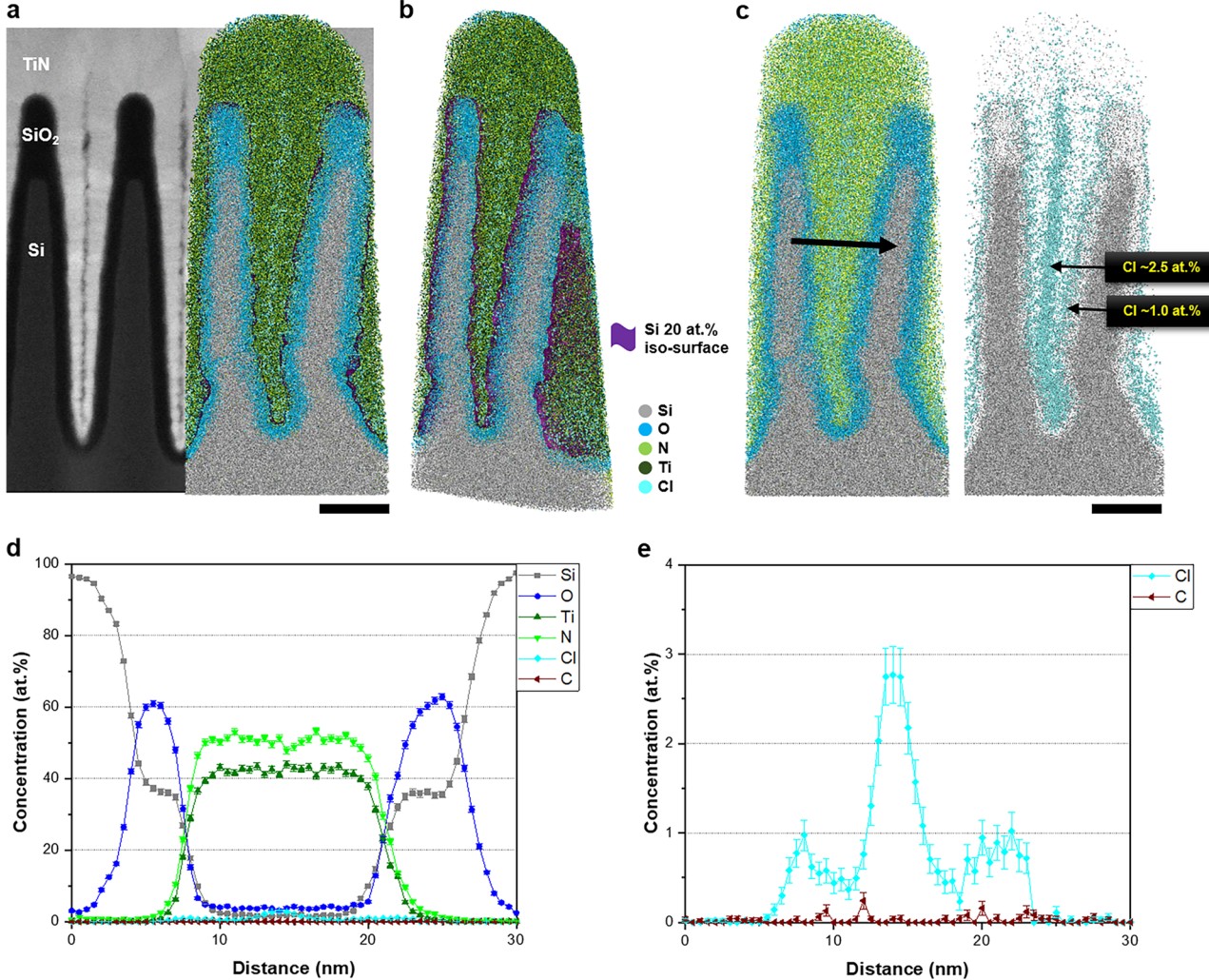

**Fig. 7 | APT analysis of the Si/SiO₂/TiN fin-structured device. a** APT 3D reconstruction overlaid on the HAADF-STEM image. **b** Reconstructed APT 3D ion map after being cropped along the $Y$–$Z$ plane to improve visualization. The Si at the SiO₂/TiN interface at an iso-concentration of 20 at.% is highlighted in purple. **c** Ion maps of N (light green), O (sky blue), Si (gray), and Cl (cyan) in the fin-structured 3D device. **d** Composition profile from one fin to another fin (indicated by the bold arrow in **c**). Some degree of elemental intermixing occurred owing to the trajectory aberration. **e** Enlarged composition profile showing the distribution of residual Cl (Scale bars: 20 nm in **a**–**c**). Error bars in (**d**, **e**) indicate 1-sigma counting statistics.

a detrimental effect on the reliability, the formation of an optimized diffusion barrier is essential.

## Application to three-dimensional device structure

We investigated the N distribution in a 3D device with a Si/SiO₂/TiN structure to estimate the conformality for N analysis in real semiconductor devices. The 3D device structure is illustrated by the reconstructed APT ion map superimposed on a STEM-HAADF image captured normal to the fin structure and cropped along the $Y$–$Z$ plane (Fig. 7a, b). The SiO₂/TiN interface was outlined using a 20 at.% iso-concentration surface to improve visualization. The layers along the vertical direction were widened and showed slight intermixing with neighboring layers owing to trajectory aberrations in the APT analysis, as well documented in previous studies[29,32,74–76] the dimensions and structure revealed by the reconstructed APT ion map showed good agreement with those observed in the STEM image. Our APT analysis enabled the direct simultaneous observation of the 3D distributions of N, Si, and SiOₓ, rather than representing N by visualizing the TiN distributions. The N₂⁺ peak was clearly distinguished from the Si⁺ peak extracted from a certain volume of the reconstructed APT ion map (Supplementary Fig. 16). The resulting N map, independent of the TiN map, provides valuable insights into the impact of the N distribution within actual 3D device structures, effectively visualizing its spatial distribution even in the presence of Si. In addition, the resulting Cl segregation at the grain boundaries of TiN and the accumulation of Cl at the SiO₂/TiN interface were clearly visible (Fig. 7c–e and Supplementary Fig. 17). Cl primarily originated from the TiCl₄ used in the deposition of TiN and readily diffused toward the grain boundaries or interface. Because the properties of semiconductor devices can be deteriorated by Cl-induced trap sites, the distribution of Cl must be monitored along with those of SiO₂ and SiON. Our analytical approach confirms that the N distribution in Si-based 3D devices can be accurately determined at the atomic scale using APT.

This study successfully characterizes the 3D N distribution in Si-based semiconductors using advanced APT, which has an extended flight length, with optimized analytical conditions. By directly analyzing N in thin (5 nm) and ultra-thin (2 nm) SiO₂/SiON gate dielectrics within real device structures such as DRAM and logic devices, we provided crucial insights into how incorporating N modulates the impurity behavior and dielectric properties. Our findings show that PN significantly influenced the N profile, with a shallow N distribution and higher N concentration in the SiON layer effectively suppressing

impurity diffusion and improving the electrical properties. Specifically, N engineering minimized its penetration into the Si/SiO$_2$ interface while creating a robust SiON layer that prevented Cl and B diffusion from the gate electrode. These findings establish APT as a powerful tool for characterizing ultra-thin gate dielectrics and complex 3D device architectures, offering a fundamental understanding of the N distribution, which is critical for the next generation of semiconductor technology.

## Methods

### Scanning transmission electron microscopy analysis

Cross-sectional specimens for STEM observation were fabricated using a Helios5 HX dual-beam focused ion beam system (Thermo Fisher Scientific). To mitigate potential damage from Ga ions, the final thinning steps were performed at 5 kV and 2 kV using low ion currents of 16 pA and 3 pA, respectively. High-resolution imaging was conducted on a 300 kV spherical aberration-corrected Titan Cubed microscope. For elemental mapping, STEM-EDS was acquired under a probe current of 70 pA over a total integration time of 600 s at a nominal magnification of 1.3 M. EELS was carried out using a Gatan Imaging Filter in spectrum imaging mode, employing a convergence semi-angle of 26 mrad and a collection semi-angle of 43 mrad. The energy dispersion was set to 0.05 eV per channel, with a pixel size of 0.23 nm and dwell times per pixel ranging from 0.2 to 1 s. A low beam current of 50 pA was maintained throughout the analysis to minimize beam-induced damage, which was evaluated by examining the specimen before and after EELS acquisition for any structural changes.

### Secondary-ion mass spectrometry analysis

SIMS analysis was conducted using an ION-TOF M6 instrument (ION-TOF GmbH, Muenster, Germany) operated in dual-beam configuration. A Bi$^+$ primary ion beam at 30 keV with a current of 1.0 pA was utilized for acquisition, while a 1 keV Cs$^+$ ion beam at 20 nA served as the sputtering source for depth profiling under negative ion detection mode. The sample was etched over an area of $300 \times 300$ μm$^2$, with analytical data collected from a central $100 \times 100$ μm$^2$ region with a resolution of $128 \times 128$ pixels. Charge compensation techniques were not employed during the measurements.

### Atom probe tomography specimen preparation

APT specimens were prepared using a site-specific lift-out approach via a Helios5 HX focused ion beam system. Before the FIB process, poly-Si or TiN—materials actually used as gate electrodes—had to be deposited on the gate dielectrics to prevent the outgassing of N. For the same reason, the exposure of the APT sample to air had to be minimized by transferring it to the APT chamber as quickly as possible. To protect the surface during specimen preparation, a two-step platinum deposition was performed over the region of interest (ROI, $12 \times 1.7$ μm): a 100 nm layer using electron beam-induced deposition followed by a 1 μm layer applied with a Ga ion beam. Afterward, the protected ROI was extracted and mounted onto a pre-sharpened tungsten tip. Needle sharpening was carried out through annular milling at 30 kV and 80 pA to form a tapered geometry, followed by low-energy cleaning at 5 kV and 8 pA to precisely thin the specimen near the target region and reduce Ga-induced damage. Because the gate dielectric layer is highly vulnerable to both electron and ion beam exposure, minimizing exposure during preparation was crucial. The samples were prepared with consistent geometry and uniform ROI placement, ensuring that the distance from the sample apex to the target ROI remained nearly identical across measurements at ~35 nm. Consequently, the analysis voltage was maintained at ~3 kV when passing through the gate dielectrics, ensuring uniform field conditions under the optimized APT settings, as shown in the voltage profiles (Supplementary Fig. 18).

### Atom probe tomography analysis and reconstruction

APT analysis was performed using a CAMECA Invizo 6000 instrument in deep-UV laser pulsing mode ($\lambda = 257.5$ nm). Operating conditions included a 200 kHz laser repetition rate, 100 pJ pulse energy, and a sample temperature of 50 K. The detection rate was set to 0.01 atoms pulse$^{-1}$. The Invizo 6000 system incorporates a dual Einzel lens configuration and an elongated flight path, which together contribute to improved peak separation capability and enable a wide field of view during analysis. APT reconstruction and data processing were performed using CAMECA AP Suite 6.3 software. During the reconstruction process, the region with a higher FWHM was selected first. Because the central region exhibited an FWHM of ~1400, this area was chosen for analysis (Supplementary Table 1). The 3D reconstruction was performed based on the shank angle, ensuring that the thickness measured from TEM was accurately reflected by adjusting only the initial tip radius.

## Data availability

The data that support the findings of this study are available from the corresponding author upon request. The data generated in this study are also provided in the Source Data file. Source data are provided with this paper.

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

## Acknowledgements

This work was supported by the Analytical Engineering Group of Samsung Advanced Institute of Technology (SAIT).

## Author contributions

B.G.C., E.L., and C.C. conceived and designed the study. J.Y.W. and Y.S.S. performed the SIMS experiments. D.J.Y. conducted the XPS measurements. J.A. and S.T.P. carried out the FIB experiments. K.L., H.A., and M.S. fabricated the samples. I.-J.R. and S.H.K. contributed to the interpretation of the APT data. B.G.C. performed the FIB, STEM, and APT measurements and wrote the manuscript. All authors discussed the results and provided feedback on the manuscript.

## Competing interests

The authors declare no competing interests.
