## [Transparent Peer Review file · Nature Communications]

Direct observation of 3D nitrogen distribution in silicon-based dielectrics using atom probe tomography

Corresponding Author: Dr Byeong Gyu Chae

Editorial Note: This file contains all reviewer reports in order by version, followed by all author rebuttals in order by version. Annotated article files provided by reviewers are not included.

Version 0:

Reviewer comments:

Reviewer #1

(Remarks to the Author)

This is a timely study where APT method development resolves a long-standing problem, for how to mass-resolve N_2^+ and Si^+ .

The paper demonstrates the strength of the metal on state-of-the-art structures for semiconductor industry.

It is worthy of publishing. in NC.

Before doing that, however:

- 1) Make test experiments where the natural isotope $^{14}N_2$ is substituted by $^{15}N_2$ gas, for definite mass separation to the ^{28}Si , which would offer firm confirmation of your method.
- 2) Explain why the column boundaries in the TiN layer seen by STEM (porous or not) are not resolved for its density variation in the APT images.
- 3) Elaborate in the Abstract and the Conclusion what comprises your new method.
- 4) Differentiate more for the writing and contents of the Abstract and Conclusions, which now are quite similar.

Reviewer #2

(Remarks to the Author)

This manuscript presents an application of atom probe tomography (APT) to investigate the three-dimensional distribution of nitrogen in silicon-based dielectrics, with implications for semiconductor device performance, particularly in DRAM technology. While the study provides interesting insights into nanoscale dopant distributions, several aspects of the methodology and analysis require further clarification and expansion to substantiate the claims and contextualize the findings within the existing body of literature. The following points should be addressed before the manuscript is considered for publication.

- It is well known that nitrogen loss is present in the APT analysis of nitride materials. Could the authors clarify whether there is a preferential loss of nitrogen ions during the analysis? If so, what measures were taken to mitigate this issue, and how might it impact the quantification accuracy?
- Please expand the discussion on advanced focused ion beam (FIB) techniques employed for APT sample preparation. A detailed account would benefit readers seeking to replicate these methods.
- Discuss the challenges associated with field evaporation and the criteria used for selecting reconstruction parameters. Additionally, what steps were taken to minimize reconstruction artifacts, and what best practices do the authors recommend?
- Elaborate on the observed correlation between chlorine (Cl) and boron (B) distributions in silicon-based semiconductors and their impact on the material's electrical properties.
- Include the PN and PNA parameters in the methods section for clarity and reproducibility.
- Provide a more detailed explanation of the nitrogen diffusion mechanism at the $SiON/SiO_2$ interface compared to other interfaces within the device structure.
- A.D. Giddings et al. ("Industrial Application of Atom Probe Tomography to Semiconductor Devices," Scripta Materialia, 2018) discuss dopant distributions (P, O, and B) in finFET structures using APT. Could the authors explicitly state how their work differs from this and other existing literature? Highlighting the novelty of the current study would strengthen its impact.
- Consider including results from complementary techniques, such as STEM-EELS, to validate and cross-check the APT findings.

- Provide insights into the topological changes at the SiON/SiO₂ interface before and after PN and PNA processes, distinguishing between shorter and longer durations.

Reviewer #3

(Remarks to the Author)

The authors provide a very impressive overview of the capabilities of the new generation of atom probes with an enhanced mass resolution that finally allows for separating N and Si they report on the N distribution in a set of carefully crafted Si-based materials, culminating with actual device structures. As a demonstration of capabilities, the paper is fantastic, the implications in terms of properties is lacking a bit – i.e. what do we truly learn from the distribution of N revealed by APT? I think this could be important to discuss more in depth, particularly for devices.

There are some wrong statements in the paper and an apparent lack of knowledge of the relevant literature. I think nothing that cannot be corrected in a revised version of this manuscript.

My main comments can be found in the attached pdf.

Version 1:

Reviewer comments:

Reviewer #1

(Remarks to the Author)

Authors have revised their manuscript satisfactory.

Reviewer #2

(Remarks to the Author)

The authors have adequately addressed the reviewer's comments and concerns. I have a few minor suggestions for improving the manuscript.

- The authors mention using SIMS, STEM-EDS, and EELS techniques in the manuscript; however, detailed experimental procedures for these methods are missing from the "Methods" section. Including these details would improve clarity and reproducibility.
- Figure 4: The micron scale marker in the STEM image is not legible and should be improved for better visualization.
- Figure 6: The STEM-HAADF image lacks a micron scale marker, which should be added.
- Figure 7: In the APT analysis of the Si/SiO₂/TiN fin-structured device, the color coding for elements (Si, O, N) appears inconsistent with earlier figures. This inconsistency may lead to confusion; a uniform color scheme across all figures is recommended for clarity.
- Supplementary Figures 10 and 11(c, d): These figures lack clarity, and the presented information is difficult to discern. Enhancing the resolution, axis labels, and contrast would significantly improve readability and the overall quality of the presentation.

Reviewer #3

(Remarks to the Author)

I am satisfied with the reply to my comments and the associated modifications to the manuscript. IMHO, the paper can now be published

RESPONSE TO REVIEWER'S COMMENTS

Reviewer #1 (Remarks to the Author):

General Comment: This is a timely study where APT method development resolves a long-standing problem, for how to mass-resolve N_2^+ and Si^+ . The paper demonstrates the strength of the metal on state-of-the-art structures for semiconductor industry. It is worthy of publishing. in NC.

Answer to General Comment: Thank you for recognizing the significance of this work. We truly appreciate your acknowledgment of the impact of our APT research in resolving this long-standing challenge. We have addressed your comments to the best of our ability and have revised the manuscript accordingly.

Before doing that, however:

Comment 1: Make test experiments where the natural isotope $14N_2$ is substituted by $15N_2$ gas, for definite mass separation to the $28Si$, which would offer firm confirmation of your method.

Answer to Comment 1: We sincerely appreciate your thoughtful suggestion and fully understand the value of conducting test experiments using $^{15}N_2$ gas to achieve definite mass separation from ^{28}Si for further confirmation of our method. We genuinely wish we could accommodate this request; however, unfortunately, it is not feasible in our case. The samples analyzed in our study were fabricated within a highly controlled semiconductor manufacturing environment (fab), where only materials and gases essential for product production are permitted. We have been informed that introducing $^{15}N_2$ gas into this process is strictly prohibited, making it impossible for us to conduct such experiments within our current constraints. Additionally, we reached out to neighboring universities and research institutions for collaboration, but they also faced challenges in fabricating such samples due to the complexity of successful isotope incorporation. If another research group is able to provide a well-adhered thin film sample containing a sufficient amount of $^{15}N_2$, we would be more than willing to analyze it and contribute to further validating our findings. We sincerely regret that we are unable to fully accommodate this request and truly apologize once again.

Despite this limitation, we have made extensive efforts to validate the reliability of our results through multiple approaches. First, we have added the APT mass spectrum from 13 to 33 Da, demonstrating the separation of N from Si. This has been included in Supplementary Fig. 2, and the TOF spectrum is additionally presented below. We also analyzed the variation in Si and N signal behavior as a function of laser pulse energy to confirm that the two peaks shift in distinct ways, further supporting our analysis results. These findings have been incorporated into the revised manuscript and Supplementary Fig. 4. Additionally, we conducted STEM-EDS and STEM-EELS analyses to provide further validation. While STEM-EDS confirmed the presence of the SiON layer in the $SiO_2/SiON$ structure, it was not suitable for comparing N content. STEM-EELS provided reliable N quantification but only under highly controlled analytical conditions. However, we observed that the $SiO_2/SiON$ structure was extremely vulnerable to e-beam exposure, leading to sample degradation and delamination, making the experiments highly challenging. Through extensive trial and error, we were able to obtain meaningful STEM-EELS results. This challenge has reinforced our belief in the necessity of 3D N distribution analysis using APT. We have included the STEM-EDS and STEM-EELS results in the revised manuscript and Supplementary Figs. 9 and 11. We hope that our extensive efforts to strengthen the

reliability of our analysis and address your concerns to the best of our ability are well understood. Thank you again for your valuable feedback.

The manuscript has been revised as follows:

Please see Page #5,

“Detecting N in Si-based materials is therefore particularly challenging, despite its necessity in semiconductor applications; however, under optimized analytical conditions, the Invivo 6000 can differentiate between Si and N ions owing to its improved mass-resolving power (Fig. 1 and Supplementary Figs. 1 and 2). Figure 1a shows a 3D atom map of a 30-nm-thick SiN layer deposited on a Si substrate. Although the mass peaks of Si^+ and N_2^+ (shown in gray and green, respectively) partially overlap, they are sufficiently separated to identify each ion (Fig. 1b and Supplementary Figs. 2 and 3). The 3D elemental maps of Si and N on the 30-nm-thick SiN are shown in Figs. 1c and 1d respectively. N was observed as both N^+ and N_2^+ ; however, for visualization, only N_2^+ was used as its peak better separates from that of Si in the APT mass spectrum. Despite having approximately the same mass-to-charge ratio, the distributions of N_2^+ and Si^+ are not correlated. As expected, N_2^+ is located in the SiN layer, while Si^+ is distributed at a relatively low density in the SiN layer and native SiO_2 layer but at a high density on the Si substrate. The variation in the mass peak intensities between Si^+ and N_2^+ across a laser pulse energy range of 25 to 100 pJ further corroborates the presence of two distinct elements (Supplementary Fig. 4). Consequently, our results demonstrate that APT can directly characterize the N distribution in Si-based materials.”

Please see Supplementary Fig. 2,

Supplementary Fig. 2 APT mass spectra from 13 to 33 Da. N can be distinguished from Si at both ~14 and ~28 Da, although their respective peaks partially overlap.

Please see Supplementary Fig. 4.

Supplementary Fig. 4 Si⁺ and N₂⁺ mass peaks as a function of the laser pulse energy. The variation in mass peak intensities is shown across a laser pulse energy range of 25 to 100 pJ. Considering both the mass peak behavior and APT success yield, 100 pJ was selected as the optimal energy for analysis.

Please see Supplementary Fig. 9.

Supplementary Fig. 9 STEM-EELS results of Si/SiO₂/SiON/TiN structure. STEM-EELS **a** profile and **b** elemental maps. A comparative analysis between samples was feasible under an extremely low

e-beam dose and short EELS acquisition time per pixel. In the upper region of the gate dielectric, the N concentration was higher after $1.5t_{PN}$ followed by t_{PNA} .

Please see Supplementary Fig. 11.

Supplementary Fig. 11 STEM-EELS and EDS results of Si/SiO₂/SiON/poly-Si structure. **a** STEM-EELS profile and **b** ROI. A comparative analysis between samples was feasible under extremely low e-beam dose and short EELS acquisition time per pixel. In the upper region of the gate dielectric, the N concentration was higher after $1.5t_{PN}$ followed by $0.15t_{PNA}$. STEM-EDS profiles and elemental maps of **c** after PN for t_{PN} followed by PNA for t_{PNA} and **d** after PN for $1.5t_{PN}$ followed by PNA for $0.15t_{PNA}$. Directly comparing the N concentration between samples is challenging owing to the limitations of

EDS in detecting N in such thin layers.

APT ToF data at $m/z \approx 28$. Two distinct mass peaks are clearly observed.

Comment 2: Explain why the column boundaries in the TiN layer seen by STEM (porous or not) are not resolved for its density variation in the APT images.

Answer to Comment 2: We appreciate your keen observation and valuable insights. The column boundaries in the TiN layer observed in STEM correspond to the regions where two layers meet during TiN growth by ALD inside the trench structure. Following your suggestion, we revisited the APT data of this region but did not observe a clear density variation (Supplementary Fig. 18). However, Cl segregation within the column boundaries remains distinct, as reconfirmed by the composition profile we added in the revised Fig. 7. Additionally, Cl segregation along grain boundaries within the TiN layer (aligned in the vertical direction to the column boundaries) was also observed.

Thus, the absence of density variation in APT is likely due to the inherent challenges in void analysis using APT. While Wang et al. recently reported on nanovoid interpretation using APT, such analyses remain highly challenging, particularly for complex 3D structures rather than bulk materials. Furthermore, the column boundaries seen in STEM as straight-line features likely result from the overlapping of slightly misaligned columnar TiN growth fronts, rather than a perfectly uniform boundary. During APT analysis, such regions are prone to trajectory aberrations, leading to intermixing effects. This intermixing could obscure potential density variations, and it may also contribute to the Cl distribution appearing slightly broader than it actually is. To further support this discussion, we have added composition profiles and sliced 3D ion maps in the revised Fig. 7 and Supplementary Fig. 18, respectively.

The manuscript has been revised as follows:

Please see Fig. 7,

Fig. 7 APT analysis of the Si/SiO₂/TiN fin-structured device. **a** APT 3D reconstruction overlaid on the HAADF-STEM image. **b** Reconstructed APT 3D ion map after being cropped along the Y–Z plane to improve visualization. The Si at the SiO₂/TiN interface at an iso-concentration of 20 at.% is highlighted in red. **c** Ion maps of N (light green), O (blue), Si (orange), and Cl (cyan) in the fin-structured 3D device. **d** Composition profile from one fin to another fin (indicated by the bold arrow in **e**). Some degree of elemental intermixing occurred owing to the trajectory aberration. **e** Enlarged composition profile showing the distribution of residual Cl.

Please see Supplementary Fig. 18.

Supplementary Fig. 18 Sliced 3D ion maps. The density variation in the porous regions of TiN was not resolved, likely owing to the trajectory aberration.

Comment 3: Elaborate in the Abstract and the Conclusion what comprises your new method.

Thank you for your valuable feedback. In response to your suggestion, we have clarified our methodology in both the Abstract and Conclusion sections. Specifically, we emphasize that our study utilizes advanced atom probe tomography (APT) under optimized analytical conditions to analyze actual device structures with high spatial resolution and quantitative accuracy. For further details on the methodology, we have revised the Methods section accordingly. The Abstract and Conclusion now briefly mention this aspect to ensure clarity while maintaining conciseness.

The manuscript has been revised as follows:

Please see Page #2,

“Abstract

The distribution of N in semiconductor devices plays a crucial role in tuning their physical and electrical properties. However, direct observation and precise quantification of N remain challenging because of analytical limitations, particularly at critical interfaces in Si-based semiconductors. Although atom probe tomography (APT) has emerged as a powerful tool, distinguishing N from Si without isotope doping has been persistently difficult. In this study, we employed advanced APT with an extended flight path under optimized conditions to characterize the three-dimensional (3D) N distribution in actual device structures, including 2- and 5-nm-thick SiO₂/SiON-based gate dielectrics and a fin-structured 3D device. Our analysis revealed that the N distribution determines the formation of the N profile in gate dielectrics, which in turn affects the diffusion of impurities, ultimately impacting the electrical properties and reliability. Our work provides unprecedented insights into atomic-scale N behavior,

paving the way for advancing next-generation semiconductor devices.”

Please see Page #14.

“Conclusions

This study successfully characterized the 3D N distribution in Si-based semiconductors using advanced APT, which has an extended flight length, with optimized analytical conditions. By directly analyzing N in thin (5 nm) and ultra-thin (2 nm) SiO₂/SiON gate dielectrics within real device structures such as DRAM and logic devices, we provided crucial insights into how incorporating N modulates the impurity behavior and dielectric properties. Our findings show that PN significantly influenced the N profile, with a shallow N distribution and higher N concentration in the SiON layer effectively suppressing impurity diffusion and improving the electrical properties. Specifically, N engineering minimized its penetration into the Si/SiO₂ interface while creating a robust SiON layer that prevented Cl and B diffusion from the gate electrode. These findings establish APT as a powerful tool for characterizing ultra-thin gate dielectrics and complex 3D device architectures, offering a fundamental understanding of the N distribution, which is critical for the next generation of semiconductor technology.”

Please see Page #15.

“Methods

APT specimen preparation

Needle-shaped specimens for APT analysis were fabricated using the FIB lift-out method (Helios5 HX, Thermo Fisher Scientific). Before the FIB process, poly-Si or TiN—materials actually used as gate electrodes—had to be deposited on the gate dielectrics to prevent the outgassing of N. For the same reason, the exposure of the APT sample to air had to be minimized by transferring it to the APT chamber as quickly as possible. For passivation, 100-nm-thick Pt and 1- μ m-thick Pt were deposited on a region of interest (ROI) measuring 12 μ m \times 1.7 μ m on the sample surface using an electron beam and a Ga-ion beam, respectively. The ROI was then transferred onto a sharpened W tip. The samples on the W tip were sharpened into a columnar shape using an annular pattern (30 kV, 80 pA); then, milling was performed using an annular pattern (5 kV, 8 pA) to thin the samples at the location of the ROI and minimize Ga ion beam damage. Because the gate dielectric layer is highly vulnerable to both electron and ion beam exposure, minimizing exposure during preparation was crucial. The samples were

prepared with consistent geometry and uniform ROI placement, ensuring that the distance from the sample apex to the target ROI remained nearly identical across measurements at approximately 35 nm. Consequently, the analysis voltage was maintained at ~3 kV when passing through the gate dielectrics, ensuring uniform field conditions under the optimized APT settings, as shown in the voltage profiles (Supplementary Fig. 19).”

Please see Page #16,

“APT analysis and reconstruction

APT analysis was performed using a CAMECA atom probe (Invivo 6000) in deep UV laser mode ($\lambda = 257.5$ nm) at a 200 kHz pulse repetition rate and base temperature of 50 K base temperature, along with a laser-pulsed energy of 100 pJ at a detection rate of 0.01 atoms/pulse detection rate. The atom probe employs double einzel lens technology with an extended flight path to provide a large field of view and high mass-resolving power. APT reconstruction and data processing were performed using CAMECA AP Suite 6.3 software. During the reconstruction process, the region with higher FWHM was selected first. Because the central region exhibited an FWHM of approximately 1400, this area was chosen for analysis (Supplementary Fig. 1). The 3D reconstruction was performed based on the shank angle, ensuring that the thickness measured from TEM was accurately reflected by adjusting only the initial tip radius.”

Comment 4: Differentiate more for the writing and contents of the Abstract and Conclusions, which now are quite similar.

Thank you for your valuable feedback. We acknowledge that the Abstract and Conclusion were somewhat similar, as you pointed out, and we have revised them accordingly to enhance their differentiation. Thank you again for your constructive suggestion.

The manuscript has been revised as follows:

Please see Page #2,

“Abstract

The distribution of N in semiconductor devices plays a crucial role in tuning their physical and electrical properties. However, direct observation and precise quantification of N remain challenging because of

analytical limitations, particularly at critical interfaces in Si-based semiconductors. Although atom probe tomography (APT) has emerged as a powerful tool, distinguishing N from Si without isotope doping has been persistently difficult. In this study, we employed advanced APT with an extended flight path under optimized conditions to characterize the three-dimensional (3D) N distribution in actual device structures, including 2- and 5-nm-thick SiO₂/SiON-based gate dielectrics and a fin-structured 3D device. Our analysis revealed that the N distribution determines the formation of the N profile in gate dielectrics, which in turn affects the diffusion of impurities, ultimately impacting the electrical properties and reliability. Our work provides unprecedented insights into atomic-scale N behavior, paving the way for advancing next-generation semiconductor devices.”

Please see Page #14.

“Conclusions

This study successfully characterized the 3D N distribution in Si-based semiconductors using advanced APT, which has an extended flight length, with optimized analytical conditions. By directly analyzing N in thin (5 nm) and ultra-thin (2 nm) SiO₂/SiON gate dielectrics within real device structures such as DRAM and logic devices, we provided crucial insights into how incorporating N modulates the impurity behavior and dielectric properties. Our findings show that PN significantly influenced the N profile, with a shallow N distribution and higher N concentration in the SiON layer effectively suppressing impurity diffusion and improving the electrical properties. Specifically, N engineering minimized its penetration into the Si/SiO₂ interface while creating a robust SiON layer that prevented Cl and B diffusion from the gate electrode. These findings establish APT as a powerful tool for characterizing ultra-thin gate dielectrics and complex 3D device architectures, offering a fundamental understanding of the N distribution, which is critical for the next generation of semiconductor technology.”

Reviewer #2 (Remarks to the Author):

General Comment: This manuscript presents an application of atom probe tomography (APT) to investigate the three-dimensional distribution of nitrogen in silicon-based dielectrics, with implications for semiconductor device performance, particularly in DRAM technology. While the study provides interesting insights into nanoscale dopant distributions, several aspects of the methodology and analysis require further clarification and expansion to substantiate the claims and contextualize the findings within the existing body of literature. The following points should be addressed before the manuscript is considered for publication.

Answer to General Comment: Thank you for recognizing the importance of this work in the application of APT for semiconductor device analysis and for your valuable comments. As you pointed out, we have made every effort to enhance the reliability of our study and clarify its relevance in comparison to existing literature. We have responded to your comments to the best of our ability and have revised the manuscript accordingly.

Comment 1: It is well known that nitrogen loss is present in the APT analysis of nitride materials. Could the authors clarify whether there is a preferential loss of nitrogen ions during the analysis? If so, what measures were taken to mitigate this issue, and how might it impact the quantification accuracy?

Answer to Comment 1: Thank you for your insightful question. It is well known that N loss can occur during APT analysis of nitride materials, primarily due to multi-hit detection effects and the presence of N₂ gas species. In our study, the total N content in the thin film is not precisely known, and our analysis is primarily comparative, meaning that we focus on relative quantification between samples with the same structure rather than absolute quantification.

Nevertheless, to assess the impact of preferential N loss, we conducted experiments examining the variation in Si⁺ and N₂⁺ peak intensities as a function of laser pulse energy (ranging from 25 pJ to 100 pJ). Our results demonstrated that increasing the laser pulse energy led to a reduction in the N₂ signal, indicating potential N loss at higher energy levels. However, achieving sufficient success yield in APT analysis is also crucial for reliable and extensive data acquisition. Therefore, we conducted our experiments under conditions that allowed for adequate N detection while maintaining a stable success yield (a laser-pulsed energy of 100 pJ at a detection rate of 0.01 atoms/pulse detection rate), ensuring reliable quantification. Furthermore, to enhance the reliability of the relative quantification of N content, we compared N concentrations as a function of CSR values. As a result, samples with different N contents exhibited clearly distinguishable quantitative trends, even at similar CSR values. We have revised the manuscript to clarify our methodology and findings.

The manuscript has been revised as follows:

Please see Page #5,

“The variation in the mass peak intensities between Si⁺ and N₂⁺ across a laser pulse energy range of 25 to 100 pJ further corroborates the presence of two distinct elements (Supplementary Fig. 4).

Consequently, our results demonstrate that APT can directly characterize the N distribution in Si-based materials.”

Please see Page #8,

“The SiON layer contained approximately 20 at.% N after a short PN time but more than 28 at.% after a longer PN time under similar charge state ratio (CSR) values (Supplementary Fig. 12). Because APT field evaporation conditions can vary depending on the analysis environment, comparing data at similar CSR values—an indicator of comparable field conditions—is preferable for ensuring a reliable compositional analysis.^{18,64-66}”

Please see Supplementary Fig. 4,

Supplementary Fig. 4 Si^+ and N_2^+ mass peaks as a function of the laser pulse energy. The variation in mass peak intensities is shown across a laser pulse energy range of 25 to 100 pJ. Considering both the mass peak behavior and APT success yield, 100 pJ was selected as the optimal energy for analysis.

Please see Supplementary Fig. 12,

Supplementary Fig. 12 N concentration as a function of the CSR value (Si²⁺/Si⁺). At similar CSR values, the trends in the N concentration differ significantly depending on the PN & PNA process.

References have been added as follows:

64. Miller, M. K. & Smith, G. D. W. An atom probe study of the anomalous field evaporation of alloys containing silicon. *J. Vac. Sci. Technol.* **19**, 57–62 (1981). [10.1116/1.571017](https://doi.org/10.1116/1.571017).
65. Shariq, A., et al. Investigations of field-evaporated end forms in voltage- and laser-pulsed atom probe tomography. *Ultramicroscopy* **109**, 472–479 (2009). [10.1016/j.ultramic.2008.10.001](https://doi.org/10.1016/j.ultramic.2008.10.001)
66. Mancini, L., et al. Composition of wide bandgap semiconductor materials and nanostructures measured by atom probe tomography and its dependence on the surface electric field. *J. Phys. Chem. C*. **118**, 24136–24151 (2014). [10.1021/jp5071264](https://doi.org/10.1021/jp5071264).

Comment 2: Please expand the discussion on advanced focused ion beam (FIB) techniques employed for APT sample preparation. A detailed account would benefit readers seeking to replicate these methods.

Answer to Comment 2: Thank you for your valuable suggestion. While we have described the FIB-based APT sample preparation in the Methods section, we recognize that additional details could improve clarity for readers attempting to replicate our approach. As per your recommendation, we have expanded the discussion accordingly. One critical aspect of the preparation process is the need for a protective layer over the SiO₂/SiON stack due to potential N outgassing. Conventional metal coatings such as Ni or Cr are often used as protective layers but may react with the SiO₂/SiON dielectric, making them unsuitable in this case. To mitigate this issue, we utilized poly-Si and TiN—materials commonly employed as actual gate electrodes—ensuring minimal interference with the SiO₂/SiON structure. For

FIB milling, we precisely targeted the SiO₂/SiON region while minimizing beam-induced damage. Initial coarse milling was performed using 30 kV, 40 pA (or 80 pA) until reaching a depth of 300–350 nm from the target region. Final fine milling was conducted at 5 kV, 8 pA or 16 pA, gradually thinning the sample until reaching approximately 35 nm from the target region. Because SiON layers are highly sensitive to electron and ion beam exposure, even low-dose e-beam or ion beam interactions can cause delamination or structural degradation. Thus, we carefully optimized the FIB parameters to minimize potential damage while ensuring high-quality sample preparation for APT analysis. These details have been incorporated into the revised Methods section to provide a more comprehensive account of our approach.

The manuscript has been revised as follows:

Please see Page #15.

“Methods

APT specimen preparation

Needle-shaped specimens for APT analysis were fabricated using the FIB lift-out method (Helios5 HX, Thermo Fisher Scientific). *Before the FIB process, poly-Si or TiN—materials actually used as gate electrodes—had to be deposited on the gate dielectrics to prevent the outgassing of N. For the same reason, the exposure of the APT sample to air had to be minimized by transferring it to the APT chamber as quickly as possible.* For passivation, 100-nm-thick Pt and 1- μ m-thick Pt were deposited on a region of interest (ROI) measuring 12 μ m \times 1.7 μ m on the sample surface using an electron beam and a Ga ion beam, respectively. The ROI was then transferred onto a sharpened W tip. The samples on the W tip were sharpened into a columnar shape using an annular pattern (30 kV, 80 pA); then, milling was performed using an annular pattern (5 kV, 8 pA) to thin the samples at the location of the ROI and minimize Ga ion beam damage. *Because the gate dielectric layer is highly vulnerable to both electron and ion beam exposure, minimizing exposure during preparation was crucial. The samples were prepared with consistent geometry and uniform ROI placement, ensuring that the distance from the sample apex to the target ROI remained nearly identical across measurements at approximately 35 nm. Consequently, the analysis voltage was maintained at \sim 3 kV when passing through the gate dielectrics, ensuring uniform field conditions under the optimized APT settings, as shown in the voltage profiles (Supplementary Fig. 19).”*

Comment 3: Discuss the challenges associated with field evaporation and the criteria used for selecting reconstruction parameters. Additionally, what steps were taken to minimize reconstruction artifacts, and what best practices do the authors recommend?

Answer to Comment 3: Thank you for your insightful question. Addressing the challenges of field evaporation and reconstruction accuracy is crucial for ensuring reliable APT analysis. In our study, the samples were prepared with consistent geometry and uniform target region-of-interest (ROI) placement, ensuring that the distance from the sample apex to the target ROI remained nearly identical across measurements. As a result, the analysis voltage was kept relatively uniform under optimized APT conditions. The analysis voltage was controlled to remain around 3 kV when passing through the gate dielectrics, maintaining uniform field conditions under optimized APT settings as shown in voltage profiles. For reconstruction, we incorporated TEM-based thickness measurements to refine parameter selection. The reconstruction was performed using a shank angle aligning with the actual specimen geometry as closely as possible. Given that mass resolution varies across the detector map, with lower resolution in the outer regions and an FWHM mass resolution of ~1400 in the central region, we selected a consistent central area for reconstruction process. We have included these details in the revised manuscript to provide additional clarity on our reconstruction methodology and best practices.

The manuscript has been revised as follows:

Please see Page #16,

“Methods

APT analysis and reconstruction

APT analysis was performed using a CAMECA atom probe (Invivo 6000) in deep UV laser mode ($\lambda = 257.5$ nm) at a 200 kHz pulse repetition rate and base temperature of 50 K base temperature, along with a laser-pulsed energy of 100 pJ at a detection rate of 0.01 atoms/pulse detection rate. The atom probe employs double einzel lens technology with an extended flight path to provide a large field of view and high mass-resolving power. APT reconstruction and data processing were performed using CAMECA AP Suite 6.3 software. During the reconstruction process, the region with higher FWHM was selected first. Because the central region exhibited an FWHM of approximately 1400, this area was chosen for analysis (Supplementary Fig. 1). The 3D reconstruction was performed based on the shank angle, ensuring that the thickness measured from TEM was accurately reflected by adjusting only the initial tip radius.”

Please see Supplementary Fig. 19,

Supplementary Fig. 19 APT voltage profiles. Each sample was analyzed under controlled voltage profiles.

Please see Supplementary Fig. 1,

For the differentiation of N from Si,

Mass difference ($\text{Si}^{2+} - \text{N}^+$) = 0.0146 Da

($\text{Si}^{2+} \approx 13.9885$ Da, $\text{N}^+ \approx 14.0031$ Da)

Mass difference ($\text{Si}^{2+} - \text{N}_2^+$) = 0.029 Da

(Si⁺ \approx 27.977 Da, N₂⁺ \approx 28.006 Da)

A mass resolution of $m/\Delta m \approx 960$ (full width at half maximum (FWHM)) is theoretically required to resolve both Si²⁺/N⁺ and Si⁺/N₂⁺. However, considering peak broadening in the experimental APT mass spectra, a higher resolution—approximately 1.5 times greater of approximately $m/\Delta m \approx 1440$ —is likely needed.

Detector X × Detector Y	Full width half maximum (FWHM)
20 mm x 20 mm	1476.21
40 mm x 40 mm	1273.24
Total	1060.43

Supplementary Fig. 1 FWHM for distinguishing N from Si and the FWHM as a function of distance from the detector center in the Invizo 6000.

Comment 4: Elaborate on the observed correlation between chlorine (Cl) and boron (B) distributions in silicon-based semiconductors and their impact on the material's electrical properties.

Answer to Comment 4: We appreciate your comment. However, our study did not specifically discuss about the direct correlation between Cl and B distributions, and to our knowledge, there is no intrinsic relationship between them. Instead, our study focused on the correlation between N distribution and the impurity diffusion, particularly regarding Cl and B independently. Although we did not discuss about the correlation between Cl and B distributions, we have expanded our discussion on their individual effects on electrical properties. We have also clarified this point in the revised manuscript to prevent any potential misinterpretation.

Our findings indicate that a higher N concentration in the SiON layer effectively suppresses Cl penetration from the TiN gate electrode into the gate dielectric in the SiO₂/SiON/TiN structure. Cl, primarily originating from TiCl₄ used in TiN deposition, tends to accumulate at the SiON/TiN interface, with minor amounts diffusing into the gate dielectric. Once inside the gate dielectric, Cl can introduce defects that act as charge injection sites, leading to time-dependent dielectric breakdown (TDDB) or negative bias temperature instability (NBTI) degradation. Additionally, Cl may react with pre-existing defects or dangling bonds at the Si/SiO₂ interface, increasing interface state density and degrading carrier mobility.

Similarly, we observed that a higher N concentration in the SiON layer effectively inhibits B diffusion from the poly-Si gate into the gate dielectric in the SiO₂/SiON/poly-Si structure. B is intentionally doped into the poly-Si gate to modulate the work function and electrical characteristics. However, B diffusion

into the gate dielectric can significantly increase gate leakage current, compromising the insulating properties and overall reliability of the device.

To mitigate these detrimental effects, precise engineering of the N profile through optimized PN is essential. By carefully tailoring the N distribution, impurity diffusion can be effectively controlled, ensuring high-performance and reliable gate dielectrics. These refinements have been incorporated into the revised manuscript to provide a clearer understanding of the individual impacts of Cl and B on electrical properties.

The manuscript has been revised as follows:

Please see Page #10,

“Cl that penetrates the gate dielectric can either form a dangling bond that attracts electrons at the interface or act as a mobile ion in the insulator, thereby inhibiting charge transfer. These impurity-induced defects can degrade device performance by disrupting charge transport, increasing leakage currents, and accelerating failure mechanisms such as time-dependent dielectric breakdown (TDDB) and negative bias temperature instability (NBTI).^{67,68} Accordingly, preventing the penetration of impurities is essential to improve their properties. The trap density and the resulting degradation in reliability can be effectively suppressed through characterization and modification of the N distribution.”

Please see Page #13,

“Although B is intentionally doped into the poly-Si gate to modulate the work function and optimize the electrical properties, it can degrade device reliability by increasing the gate leakage current if it diffuses into the gate dielectric. Our analytical approach confirmed the importance of a robust SiON layer with a high N concentration to inhibit the diffusion of dopants and impurities. Because diffusion from the gate often has a detrimental effect on the reliability, the formation of an optimized diffusion barrier is essential.”

Comment 5: Include the PN and PNA parameters in the methods section for clarity and reproducibility.

Answer to Comment 5: We apologize for not initially providing detailed PN and PNA parameters and appreciate your suggestion to include them for clarity and reproducibility. While we understand the importance of providing detailed processing conditions, we are unable to disclose the exact PN and PNA parameters due to confidentiality concerns related to product development. However, we can confirm that the PN and PNA durations are on the order of a few seconds, which aligns with typical process conditions. We have ensured that our manuscript focuses on the key scientific contribution of

this work—demonstrating the feasibility of analyzing N distribution and concentration in actual semiconductor materials and devices using an advanced APT, without requiring isotope doping. We appreciate your understanding and hope that our constraints regarding the disclosure of specific parameters are taken into consideration.

Comment 6: Provide a more detailed explanation of the nitrogen diffusion mechanism at the SiON/SiO₂ interface compared to other interfaces within the device structure.

Answer to Comment 6: We appreciate your valuable comment and recognize the importance of providing a more detailed explanation of N diffusion at the SiON/SiO₂ interface compared to other interfaces within the device structure. At the SiON/TiN interface, a sharp N profile is observed due to the high N content in TiN, which inherently incorporates N during deposition. In contrast, at the Si/SiO₂ interface, only a small amount of N reaches this region, primarily due to diffusion from the SiON layer during post-nitridation annealing (PNA). However, in our results, a significant amount of N was observed at the Si/SiO₂ interface, which can be attributed to the TiN annealing process. N at this interface can induce positive fixed charges, which may degrade electrical properties.

However, N diffusion at the SiO₂/SiON interface exhibits distinct behavior compared to other interfaces in the device structure. During plasma nitridation (PN), N primarily reacts with SiO₂, incorporating through O substitution or interstitial diffusion within the amorphous SiO₂ network. Due to the strong Si–O bonding, N incorporation into the deeper SiO₂ region is significantly limited. Consequently, N concentration is highest at the upper part of the SiO₂ layer, forming a gradual N profile with a diffusion tail extending into the SiO₂ layer, leading to the formation of a SiON layer rather than deep penetration. The PNA process, which provides a thermal budget, further facilitates N diffusion by enhancing atomic mobility, resulting in a slight increase in N penetration depth within SiO₂. Ultimately, N diffusion at the SiO₂/SiON interface can be controlled by adjusting the PN dose, PNA duration, and ambient atmosphere. We have incorporated a more detailed discussion of this mechanism in the revised manuscript to ensure clarity and completeness.

The manuscript has been revised as follows:

Please see Page #7,

“The tendency of N to accumulate at the Si/SiO₂ interface was attributed to the TiN annealing process rather than the PN process. Without the TiN annealing process, only a small amount of N would reach this region through thermal diffusion during PNA.”

Please see Page #9,

“We expected that a longer PN time would incorporate more N into the gate dielectric, as N primarily interacts with SiO₂ during PN through O substitution or interstitial diffusion. However, because of the strong Si–O bonding, N diffusion into the deeper SiO₂ region was significantly restricted, as shown in the N profile. As a result, the N concentration remained highest near the upper part of the SiO₂ layer,

creating a gradual N profile with a diffusion tail extending into the SiO₂. The thermal budget provided by the PNA process enhanced atomic mobility, promoting N diffusion and enabling N to penetrate slightly deeper into the SiO₂.”

Comment 7: A.D. Giddings et al. ("Industrial Application of Atom Probe Tomography to Semiconductor Devices," Scripta Materialia, 2018) discuss dopant distributions (P, O, and B) in finFET structures using APT. Could the authors explicitly state how their work differs from this and other existing literature? Highlighting the novelty of the current study would strengthen its impact.

Answer to Comment 7: We appreciate your valuable feedback and the opportunity to clarify the novelty of our study in relation to existing literature. Previous works, including the study by Giddings et al. ("Industrial Application of Atom Probe Tomography to Semiconductor Devices," Scripta Materialia, 2018), have demonstrated the capability of APT in analyzing dopant distributions (P, O, and B) in finFET structures, reinforcing its role as a powerful technique for semiconductor device analysis. However, the primary focus of our study is not the observation of dopant distribution in fin structures. Instead, we emphasize the observation of the N distribution within thin and ultra-thin gate dielectric structures on the behavior of dopants and residual impurities in actual device materials. Our findings provide direct evidence of how N incorporation influences impurity diffusion and retention in SiO₂/SiON-based gate dielectrics—a critical aspect that has not been explicitly addressed in previous APT studies. Additionally, we demonstrate that APT can effectively differentiate N from Si in fin structures, further supporting its applicability in analyzing N incorporation at the atomic scale. Notably, instead of representing N through the visualization of TiN or SiN distributions, we show that N can be independently visualized alongside the direct distribution of Si.

Our study highlights the potential of APT to elucidate the correlation between the N profile formed during PN processing and the distribution of dopants and other impurities, offering new insights into how N engineering can optimize gate dielectric properties. We recognize that these distinctions may not have been explicitly stated in the original manuscript. To address this, we have revised the manuscript to better articulate the novelty and contributions of our work.

The manuscript has been revised as follows:

Please see Page #13,

“Our APT analysis enabled the direct simultaneous observation of the 3D distributions of N, Si, and SiO_x, rather than representing N by visualizing the TiN distributions. The N₂⁺ peak was clearly distinguished from the Si⁺ peak extracted from a certain volume of the reconstructed APT ion map (Supplementary Fig. 17). The resulting N map, independent of the TiN map, provides valuable insights into the impact of the N distribution within actual 3D device structures, effectively visualizing its spatial distribution even in the presence of Si. In addition, the resulting Cl segregation at the grain boundaries

of TiN and the accumulation of Cl at the SiO₂/TiN interface were clearly visible (Figs. 7c–e and Supplementary Fig. 18).”

Please see Fig. 7.

Fig. 7 APT analysis of the Si/SiO₂/TiN fin-structured device. a APT 3D reconstruction overlaid on the HAADF-STEM image. **b** Reconstructed APT 3D ion map after being cropped along the Y–Z plane to improve visualization. The Si at the SiO₂/TiN interface at an iso-concentration of 20 at.% is highlighted in red. **c** Ion maps of N (light green), O (blue), Si (orange), and Cl (cyan) in the fin-structured 3D device. **d** Composition profile from one fin to another fin (indicated by the bold arrow in **c**). Some degree of elemental intermixing occurred owing to the trajectory aberration. **e** Enlarged composition profile showing the distribution of residual Cl.

References have been added as follows:

27. Giddings, A. D. et al. Industrial application of atom probe tomography to semiconductor devices. *Scr. Mater.* **148**, 82–90 (2018). [10.1016/j.scriptamat.2017.09.004](https://doi.org/10.1016/j.scriptamat.2017.09.004).

Comment 8: Consider including results from complementary techniques, such as STEM-EELS, to validate and cross-check the APT findings.

Answer to Comment 8: We appreciate your suggestion to include results from complementary techniques to validate our APT findings. In the Supplementary Figures, we had already included SIMS and XPS results, which we believe are in good agreement with our APT data. However, as per your request, we have additionally conducted STEM-EDS and STEM-EELS analyses to further support our findings.

Because the SiO_2/SiON structure formed via the PN process is highly vulnerable to e-beam exposure, we carefully optimized the analysis conditions to minimize beam-induced modifications. Unfortunately, STEM-EDS was not suitable for reliably comparing N content across samples, especially in the extremely thin SiO_2/SiON structure, where detecting N with sufficient signal strength was challenging. Under optimized conditions, STEM-EELS analysis yielded results consistent with those from APT and other complementary techniques. However, due to the fragility of the SiO_2/SiON sample to e-beam, EELS measurements had to be conducted under very mild analytical conditions (extremely low e-beam dose and short acquisition time per pixel), leading to relatively weak signal intensity. Additionally, e-beam exposure caused degradation and delamination of the SiON layer, making it particularly challenging to establish suitable conditions for analysis. Ensuring that the SiO_2/SiON sample remained intact required extensive time and effort, given the difficulties involved. Because analyzing N alongside other impurities with sufficient signal intensity was particularly challenging, we have included only the N and O data obtained from STEM-EELS. We appreciate your understanding of these limitations. These kinds of challenges have reinforced our belief in the necessity of 3D N distribution analysis using APT. We have revised the manuscript accordingly to clarify these details and included STEM-EDS and STEM-EELS data when discussing each structure.

The manuscript has been revised as follows:

Please see Supplementary Fig. 9,

Supplementary Fig. 9 STEM-EELS results of Si/SiO₂/SiON/TiN structure. STEM-EELS a profile

and **b** elemental maps. A comparative analysis between samples was feasible under an extremely low e-beam dose and short EELS acquisition time per pixel. In the upper region of the gate dielectric, the N concentration was higher after $1.5t_{PN}$ followed by t_{PNA} .

Please see Supplementary Fig. 11.

Supplementary Fig. 11 STEM-EELS and EDS results of Si/SiO₂/SiON/poly-Si structure. a STEM-EELS profile and **b** ROI. A comparative analysis between samples was feasible under extremely low e-beam dose and short EELS acquisition time per pixel. In the upper region of the gate dielectric, the N concentration was higher after $1.5t_{PN}$ followed by $0.15t_{PNA}$. STEM-EDS profiles and elemental maps of **c** after PN for t_{PN} followed by PNA for t_{PNA} and **d** after PN for $1.5t_{PN}$ followed by PNA for $0.15t_{PNA}$.

Directly comparing the N concentration between samples is challenging owing to the limitations of EDS in detecting N in such thin layers.

Comment 9: Provide insights into the topological changes at the SiON/SiO₂ interface before and after PN and PNA processes, distinguishing between shorter and longer durations.

Answer to Comment 9: We appreciate your question regarding potential topological changes at the SiON/SiO₂ interface before and after the PN and PNA processes. However, based on our analysis, we did not observe any significant morphological or structural differences at the interface. Both shorter and longer PN/PNA durations resulted in a similar interface profile, with no noticeable variations in roughness, thickness, or continuity of the SiON layer as shown in our STEM results. While N incorporation levels varied, leading to differences in the N depth profile and concentration, the overall topology remained consistent before and after processing. From a compositional perspective, differences in N incorporation and diffusion were observed depending on the PN and PNA conditions. With longer PNA durations, a higher N content diffused into SiO₂, resulting in a broader N profile.

Reviewer #3 (Remarks to the Author):

General Comment: The authors provide a very impressive overview of the capabilities of the new generation of atom probes with an enhanced mass resolution that finally allows for separating N and Si. They report on the N distribution in a set of carefully crafted Si-based materials, culminating with actual device structures. As a demonstration of capabilities, the paper is fantastic, the implications in terms of properties is lacking a bit – i.e. what do we truly learn from the distribution of N revealed by APT? I think this could be important to discuss more in depth, particularly for devices. There are some wrong statements in the paper and an apparent lack of knowledge of the relevant literature. I think nothing that cannot be corrected in a revised version of this manuscript. My main comments can be found in the attached pdf.

Answer to General Comment: We greatly appreciate your recognition of the significance of our research and the efforts involved, as well as your detailed review aimed at further improving this manuscript. Following your advice, we have expanded the discussion to strengthen the significance of our findings. We have addressed your comments point by point to the best of our ability and have revised the manuscript accordingly.

Comment 1: That first paragraph is too long. Hard to read.

Answer to Comment 1: We appreciate your comments aimed at improving our manuscript. We also recognized that the first paragraph was somewhat lengthy. Taking your suggestion into account, we have revised it to be more concise and clearer. These changes have been reflected in the revised manuscript.

The manuscript has been revised as follows:

Please see Page #3,

“N has been widely incorporated into high-performance semiconductor devices as its distribution and concentration can reliably modulate their physical and electrical properties. As a result, extensive research has focused on manipulating the N behavior in Si-based semiconductors, including its effects on band gap engineering,¹⁻⁴ trap density control,⁵⁻⁸ and passivation enhancement.⁹⁻¹¹ Understanding these effects requires directly observing the N distribution; however, conventional analytical techniques face significant limitations in providing accurate three-dimensional (3D) quantification.

Although transmission electron microscopy (TEM) has been used to investigate the behavior of N, especially in gate dielectrics formed by plasma nitridation (PN), it has limited capability in observing and quantifying light elements, such as N and O, especially at low concentrations.¹²⁻¹⁷ Secondary-ion mass spectrometry (SIMS) is an important reference tool for estimating N profiles; however, although SIMS provides extremely low-level detection, it only provides a 1D depth profile.¹⁸⁻²⁰ These constraints

hinder a comprehensive understanding of how N influences device performance, emphasizing the need for advanced analytical methods. In this study, we employed atom probe tomography (APT), which provides quantitative 3D compositional information with atomic-scale spatial resolution.²¹⁻²³ APT has been extensively applied in diverse areas, including semiconductor and optoelectronic devices, dielectric materials, and nanowires.²⁴⁻³⁴ However, the overlaps between the S and N mass peaks in APT mass spectra have typically made it difficult to accurately characterize the behavior of N in Si-based semiconductors.³⁵⁻³⁹ Some studies have tracked N behavior using isotopically labeled dopants such as ¹⁵N or ³⁰Si, but this approach does not accurately represent its behavior in actual device structures.^{36,40} To overcome this issue, we utilized the advanced Invivo 6000 atom probe, which features a multiple Einzel lens system and a longer flight path length than prior instruments. To the best of our knowledge, this is the first study that analyzes N in Si-based thin films and device structures using APT without isotopic doping.”

Comment 2: This is strictly speaking not true. There are thresholds beyond which they can be measured through spectroscopic methods, including EELS or EDS. I think this statement should be toned down a bit.

Answer to Comment 2: We apologize for any confusion caused by our initial explanation. We fully agree with the reviewer that the statement needed to be adjusted for accuracy. As per your suggestion, we have revised the sentence in the manuscript to clarify that TEM does not entirely preclude the observation and quantification of light elements but presents significant challenges, particularly at low concentrations, making direct comparisons between samples difficult.

The manuscript has been revised as follows:

Please see Page #3,

“Although transmission electron microscopy (TEM) has been used to investigate the behavior of N, especially in gate dielectrics formed by plasma nitridation (PN), it has limited capability in observing and quantifying light elements, such as N and O, especially at low concentrations.^{12-17”}

Comment 3: Isn't depth profiling 1D ??

Answer to Comment 3: We apologize for this mistake. You are absolutely correct that SIMS provides a 1D depth profile rather than a 2D depth profile. We have corrected this in the revised manuscript,

changing “2D depth profile” to “1D depth profile” to accurately reflect the nature of SIMS analysis. Thank you for pointing this out.

The manuscript has been revised as follows:

Please see Page #3,

“Secondary-ion mass spectrometry (SIMS) is an important reference tool for estimating N profiles; however, although SIMS provides extremely low-level detection, it only provides a 1D depth profile.¹⁸⁻

20”

Comment 4: This may not be totally true - there had been reports of isotopically doping ^{15}N or ^{30}Si for instance to measure the relative distribution of one vs. the other

in Fe: <https://www.sciencedirect.com/science/article/pii/S03960289291055G>

<https://www.sciencedirect.com/science/article/pii/S1359645421001543>

But also in Si:

<https://www.sciencedirect.com/science/article/pii/S0169433215010661>

<https://academic.oup.com/mam/articlepdf/14/S2/1230/48233304/mam1230.pdf>

Answer to Comment 4: We greatly appreciate your recognition and detailed advice. As you pointed out, previous studies have utilized isotopically labeled dopants such as ^{15}N or ^{30}Si to analyze N distribution. In our study, however, we aimed to demonstrate the direct observation and quantification of N in real Si-based thin films and device structures without artificial isotope doping. This approach allows us to better assess the actual behavior of N in semiconductor materials. To reflect this distinction, we have revised the manuscript to provide a more precise explanation.

The manuscript has been revised as follows:

Please see Page #3,

“However, the overlaps between the S and N mass peaks in APT mass spectra have typically made it difficult to accurately characterize the behavior of N in Si-based semiconductors.³⁵⁻³⁹ Some studies have tracked N behavior using isotopically labeled dopants such as ^{15}N or ^{30}Si , but this approach does not accurately represent its behavior in actual device structures.^{36,40} To overcome this issue, we utilized the advanced Invivo 6000 atom probe, which features a multiple Einzel lens system and a longer flight path length than prior instruments. To the best of our knowledge, this is the first study that analyzes N in Si-based thin films and device structures using APT without isotopic doping.”

References have been added as follows:

36. Kinno, T., Kitamoto, K., Takeno, S. & Tomita, M. Laser-assisted atom probe tomography of ^{15}N -enriched nitride thin films for analysis of nitrogen distribution in silicon-based structure. *Appl.*

Surf. Sci. **349**, 89–92 (2015). [10.1016/j.apsusc.2015.04.200](https://doi.org/10.1016/j.apsusc.2015.04.200).

40. Ulfig, R. M., Prosa, T., Reinhard, D., Oltman, E. & Alvis, R. Towards quantitative analysis of nitrogen in microelectronics applications for atom probe tomography. *Microsc. Microanal.* **14**, 1230–1231 (2008). [10.1017/S1431927608088302](https://doi.org/10.1017/S1431927608088302).

Comment 5: I think this statement is wrong. Nitrogen is detected. The issue is the discrimination between N^+ and Si^{2+} . This needs to be rephrased

Answer to Comment 5: We thank the reviewer for this helpful comment on ensuring accuracy in our explanation. As you pointed out, N can indeed be detected in APT, but the challenge lies in differentiating N^+ from Si^{2+} in the mass spectrum without the use of isotopes such as ^{15}N or ^{30}Si . To clarify this, we have revised the sentence to accurately reflect this limitation. The updated statement now emphasizes the difficulty in distinguishing N from Si in Si-based materials due to peak overlap, rather than incorrectly suggesting that APT is incapable of detecting N. This revision has been incorporated into the revised manuscript.

The manuscript has been revised as follows:

Please see Page #4,

“Conventional APT, such as the local electrode atom probe (LEAP), has difficulty distinguishing N from Si in Si-based materials because of the peak overlap in the mass spectrum, especially in the absence of isotope doping.”

Comment 6: Please quote the mass difference between N^+ and Si^{2+} and the target mass resolution allowing for discrimination. Would be good to be precise and specific.

Answer to Comment 6: We agree with the reviewer's suggestion to provide specific mass differences and the required resolution for peak discrimination. Below are the precise mass values and calculations:

$Si^{2+} \approx 13.9885$ Da

$N^+ \approx 14.0031$ Da

Mass difference ($Si^{2+} - N^+$) = 0.0146 Da

$Si^+ \approx 27.977$ Da

$N_2^+ \approx 28.006$ Da

Mass difference ($Si^{2+} - N_2^+$) = 0.029 Da

For effective discrimination, a mass resolution of $m/\Delta m \approx 960$ (FWHM) is theoretically required to resolve both Si^{2+}/N^+ and Si^+/N_2^+ . However, due to peak broadening in experimental APT mass spectra, achieving a practical separation typically requires a resolution of approximately $m/\Delta m \approx 1440$. In our study, we utilized the Invivo 6000, which provides improved mass resolution, particularly in the central region of the detector. Experimentally, we confirmed that the mass resolution in the optimized detection region was sufficient for distinguishing N from Si in Si-based materials. We have incorporated this additional information into the revised manuscript for clarity and precision.

The manuscript has been revised as follows:

Please see Page #5.

“To distinguish N from Si without isotope doping, a mass resolution of approximately $m/\Delta m \approx 1400$ is required in APT, considering the peak broadening effects inherent to the technique (Supplementary Fig. 1). Detecting N in Si-based materials is therefore particularly challenging, despite its necessity in semiconductor applications; however, under optimized analytical conditions, the Invizo 6000 can differentiate between Si and N ions owing to its improved mass-resolving power (Fig. 1 and Supplementary Figs. 1 and 2).”

Please see Supplementary Fig. 1.

For the differentiation of N from Si,

Mass difference ($\text{Si}^{2+} - \text{N}^+$) = 0.0146 Da

($\text{Si}^{2+} \approx 13.9885$ Da, $\text{N}^+ \approx 14.0031$ Da)

Mass difference ($\text{Si}^{2+} - \text{N}_2^+$) = 0.029 Da

($\text{Si}^+ \approx 27.977$ Da, $\text{N}_2^+ \approx 28.006$ Da)

A mass resolution of $m/\Delta m \approx 960$ (full width at half maximum (FWHM)) is theoretically required to resolve both $\text{Si}^{2+}/\text{N}^+$ and Si^+/N_2^+ . However, considering peak broadening in the experimental APT mass spectra, a higher resolution—approximately 1.5 times greater of approximately $m/\Delta m \approx 1440$ —is likely needed.

Detector X × Detector Y	Full width half maximum (FWHM)
20 mm x 20 mm	1476.21
40 mm x 40 mm	1273.24
Total	1060.43

Supplementary Fig. 1 FWHM for distinguishing N from Si and the FWHM as a function of distance from the detector center in the Invizo 6000.

Comment 7: Please provide a full mass spectrum, showing both the 1+ and 2+ charge states.

Answer to Comment 7: We agree your suggestion to provide a full mass spectrum, including both 1+ and 2+ charge states, to enhance clarity. In response, we have included a mass spectrum covering the 14–28 Da range, where the separation of relevant peaks can be observed. This demonstrates that N^+ and N_2^+ can be distinguished from Si^{2+} and Si^+ , respectively, under optimized conditions. The updated mass spectrum has been added to Supplementary Fig. 2, and we have revised the manuscript accordingly to reflect this additional data.

The manuscript has been revised as follows:

Please see Page #5,

“Detecting N in Si-based materials is therefore particularly challenging, despite its necessity in semiconductor applications; however, under optimized analytical conditions, the Invizo 6000 can differentiate between Si and N ions owing to its improved mass-resolving power (Fig. 1 and Supplementary Figs. 1 and 2). Figure 1a shows a 3D atom map of a 30-nm-thick SiN layer deposited on a Si substrate. Although the mass peaks of Si^+ and N_2^+ (shown in gray and green, respectively) partially overlap, they are sufficiently separated to identify each ion (Fig. 1b and Supplementary Figs. 2 and 3).”

Please see Supplementary Fig. 2,

Supplementary Fig. 2 APT mass spectra from 13 to 33 Da. N can be distinguished from Si at both ~14 and ~28 Da, although their respective peaks partially overlap.

Comment 8: Peaks

Answer to Comment 8: We thank the reviewer for this helpful comment on the use of academically appropriate terminology. We have corrected "spectra" to "peaks" in the revised manuscript to ensure accuracy and consistency.

The manuscript has been revised as follows:

Please see Page #5.

“Although the mass peaks of Si⁺ and N₂⁺ (shown in gray and green, respectively) partially overlap, they are sufficiently separated to identify each ion (Fig. 1b and Supplementary Figs. 2 and 3).”

Comment 9: Separated

Answer to Comment 9: We thank the reviewer for this helpful comment on the use of academically appropriate terminology. We have corrected "separate" to "separated" in the revised manuscript to ensure accuracy and consistency.

The manuscript has been revised as follows:

Please see Page #5.

“Although the mass peaks of Si⁺ and N₂⁺ (shown in gray and green, respectively) partially overlap, they are sufficiently separated to identify each ion (Fig. 1b and Supplementary Figs. 2 and 3).”

Comment 10: The inset is way too small, impossible to see anything

Answer to Comment 10: We agree with the reviewer's comment. However, we decided to remove the inset as we believe it is unnecessary to include. Many researchers are already aware that distinguishing N from Si without isotope doping using conventional APT systems is challenging. If necessary, we would be happy to include it again.

Comment 11: The labelling here is wrong - it should be N⁺

Answer to Comment 11: For the clarity of the meaning, we have revised the manuscript

The manuscript has been revised as follows:

Please see Page #5.

“The 3D elemental maps of Si and N on the 30-nm-thick SiN are shown in Figs. 1c and 1d respectively. N was observed as both N⁺ and N₂⁺; however, for visualization, only N₂⁺ was used as its peak better separates from that of Si in the APT mass spectrum.”

Comment 12: I think here something needs to be discussed is the reason why nitrogen would field evaporate exclusively as N_2^+ ? Kinno et al did not report a similar behaviour: <https://www.sciencedirect.com/science/article/pii/S0169433215010661> They also report $15N^+$.

Answer to Comment 12: We apologize for any confusion caused by our explanation regarding N^+ and N_2^+ field evaporation. In our study, N was observed to field evaporate in both N^+ and N_2^+ forms, consistent with the findings reported by Kinno et al. We chose to visualize N_2^+ rather than N^+ in the manuscript because N_2^+ and Si^+ exhibit better separation in the APT mass spectrum compared to N^+ and Si^{2+} , making it more suitable for clear data interpretation. To address this concern, we have shown an additional N map based on N^+ .

N map generated using only N^+ signals.

Comment 13: I don't think that's technically correct. It's near ~ 2 but not exactly 2.

Answer to Comment 13: We thank the reviewer for the helpful comment. To ensure academic accuracy, we have revised the statement accordingly. As you pointed out, the deviation from the ideal square dependence arises from factors such as multiple scattering effects and atomic arrangement.

The manuscript has been revised as follows:

Please see Page #6,

“In the magnified high-angle annular dark-field STEM (HAADF-STEM) images, in which the intensity is approximately proportional to the square of the atomic number, the SiON layer appears somewhat lighter than the SiO_2 layer (Fig. 2b);”

Comment 14: What was done?? EELS? EDX??

Answer to Comment 14: We appreciate the reviewer's comment and recognize the need to clarify the analysis techniques used. We should have provided both EDS and EELS experiment results in the manuscript. Because the SiO_2/SiON structure formed via the PN process, which is only a few nanometers thick, is highly vulnerable to electron beam exposure, it required careful experimental conditions. Under optimized conditions (extremely low electron beam dose and short acquisition time per pixel), EELS experiments sometimes provided consistent results in detecting N. However, EDS was less effective in detecting N in such a thin SiO_2/SiON structure, making it challenging to compare N concentrations across samples. These challenges have reinforced our belief in the necessity of 3D N distribution analysis using APT. We have revised the manuscript accordingly to clarify these details and included STEM-EDS and STEM-EELS data when discussing each structure.

The manuscript has been revised as follows:

Please see Supplementary Fig. 9.

Supplementary Fig. 9 STEM-EELS results of Si/SiO₂/SiON/TiN structure. STEM-EELS **a** profile and **b** elemental maps. A comparative analysis between samples was feasible under an extremely low e-beam dose and short EELS acquisition time per pixel. In the upper region of the gate dielectric, the N concentration was higher after $1.5t_{PN}$ followed by t_{PNA} .

Please see Supplementary Fig. 11.

Supplementary Fig. 11 STEM-EELS and EDS results of Si/SiO₂/SiON/poly-Si structure. **a** STEM-EELS profile and **b** ROI. A comparative analysis between samples was feasible under extremely low e-beam dose and short EELS acquisition time per pixel. In the upper region of the gate dielectric, the N concentration was higher after $1.5t_{PN}$ followed by $0.15t_{PNA}$. STEM-EDS profiles and elemental maps of **c** after PN for t_{PN} followed by PNA for t_{PNA} and **d** after PN for $1.5t_{PN}$ followed by PNA for $0.15t_{PNA}$. Directly comparing the N concentration between samples is challenging owing to the limitations of EDS in detecting N in such thin layers.

Comment 15: If not, what is the point of mentioning this??

Answer to Comment 15: Considering the points you previously mentioned, it seems appropriate to remove this sentence. We have removed the sentence.

Comment 16: I think should you rotate the datasets by 90 degrees to be in a similar orientation at the TEM.

Answer to Comment 16: Per your suggestion, we have rotated the 3D ion maps by 90 degrees to align with the TEM orientation. This revision has been incorporated into the updated manuscript.

The manuscript has been revised as follows:

Please see Fig. 3.

Fig. 3 APT analysis of the Si/SiO₂/SiON/TiN structure for DRAM applications. **a** 3D ion maps and **b** 2D composition profile of the sample without PN (left), after PN for t_{PN} followed by PNA for t_{PNA} (middle), and after PN for $1.5t_{PN}$ followed by PNA for t_{PNA} (right). The N incorporated by PN is clearly visible at the SiO₂/TiN interface in both the 3D ion maps and the 2D composition profiles. The N concentration at the interface increases with the PN time. The accumulation of N at the Si sub/SiO₂ interface is attributed to N diffusion from the TiN gate electrode during annealing.

Comment 17: What is shown here?? Is this for N_2^+ or N^+ ? Or both?? Is this just a cylinder in the middle of the reconstructed data?? Why so?

Answer to Comment 17: We appreciate the reviewer's question and have clarified these points in the revised manuscript. The N maps shown in Fig. 3a are derived from both N_2^+ and N^+ . To ensure clarity, we have explicitly stated this in the text. Regarding the 3D ion maps in Fig. 3a, as the reviewer noted, they are cylinders in the middle of the reconstructed data. This is because the outer region of the APT dataset is significantly distorted, forming a bent shape that does not correlate with the TEM data. Additionally, the Invivo 6000 provides superior mass resolution in the central region, making it more reliable for N detection. To further clarify this, we have added the full 3D reconstructed maps in Supplementary Fig. 5 and updated the manuscript accordingly.

The manuscript has been revised as follows:

Please see Page #7,

“Accordingly, the 3D distribution of N and the resulting distribution of impurities in the $SiO_2/SiON/TiN$ structure were characterized using APT. The reconstructed 3D atom maps of the $SiO_2/SiON/TiN$ structure under different PN operating parameters reveal the formation of a SiON layer on the SiO_2 layer after PN (Fig. 3a and Supplementary Fig. 5). N, which is detected as both N^+ and N_2^+ in APT, was primarily located between the SiO_2 gate dielectric and the TiN metal gate in the upper part of the 5-nm-thick gate dielectric (Fig. 3a).”

Please see Supplementary Fig. 5,

Supplementary Fig. 5 Full reconstruction maps of the Si/SiO₂/SiON/TiN structure.

Comment 18: Best stick to Ti since all the other elements are referred to by their symbol.

Answer to Comment 18: We apologize for any inconsistency in element notation. We have changed "titanium" to "Ti" in the revised manuscript for consistency with other element symbols.

Comment 19: See point above - be consistent with N or nitrogen.

Answer to Comment 19: We apologize for any inconsistency in element notation. We have changed "nitrogen" to "N" in the revised manuscript for consistency with other element symbols.

Comment 20: Ok but how does this affect quantification? I think this could be discussed more precisely. From the literature, there are also discussions of changes in the analysis conditions to change the N_2^+/N^+ ratio Was this envisaged?

Answer to Comment 20: We appreciate your insightful comment regarding the impact of N_2^+ -only quantification on accuracy and the potential influence of analysis conditions on the N_2^+/N^+ ratio. As you pointed out, relying solely on N_2^+ for quantification results in a reduced detected N signal, which can affect absolute quantification. However, in our study, the total N content in the thin film is not precisely known, and our analysis exhibits a comparative aspect to some extent, meaning that we focus on relative quantification between samples with the same structure rather than absolute quantification. Additionally, significant differences in the mass spectra between Si/SiO₂/SiON/TiN and Si/SiO₂/SiON/poly-Si structures require different mass range files (.rrng) for analysis. As noted in the manuscript, the Si/SiO₂/SiON/TiN structure contains a high N signal originating from TiN, making it necessary to apply a different mass range file for accurate quantification. Ultimately, given the current mass resolution, complete separation between Si and N is still not possible, and partial peak overlap remains. As a result, a universal quantification approach applicable to all samples remains challenging until now.

To address your concern regarding changes in analysis conditions and their effect on the N_2^+/N^+ ratio, we have added results using laser pulse energies of 25 pJ, 50 pJ, and 100 pJ. The results clearly show a reduction in the N_2^+ mass peak with increasing laser pulse energy, as demonstrated in Supplementary Fig. 4. Our main APT experiments were conducted at 100 pJ laser pulse energy, which was selected to maximize APT success yield while maintaining sufficient N signal detection. This additional information has been incorporated into the revised manuscript and supplementary Fig. 4.

The manuscript has been revised as follows:

Please see Page #5,

“Despite having approximately the same mass-to-charge ratio, the distributions of N_2^+ and Si^+ are not correlated. As expected, N_2^+ is located in the SiN layer, while Si^+ is distributed at a relatively low density in the SiN layer and native SiO₂ layer but at a high density on the Si substrate. The variation in the mass peak intensities between \$Si^+\$ and \$N_2^+\$ across a laser pulse energy range of 25 to 100 pJ further corroborates the presence of two distinct elements (Supplementary Fig. 4). Consequently, our results demonstrate that APT can directly characterize the N distribution in Si-based materials.”

Please see Supplementary Fig. 4,

Supplementary Fig. 4 Si^+ and N_2^+ mass peaks as a function of the laser pulse energy. The variation in mass peak intensities is shown across a laser pulse energy range of 25 to 100 pJ. Considering both the mass peak behavior and APT success yield, 100 pJ was selected as the optimal energy for analysis.

Comment 21: I think these comparisons can really only be made if the field conditions are comparable. This is a known issue in APT since 45y. There are numerous reports of compositional biases vs. Field since Miller's study on FeSi in 1981. This is particularly problematic for nitrides and oxides: <https://pubs.acs.org/doi/full/10.1021/jp5071264> For instance, I think the field conditions here could be assessed based on the states for instance – see <https://www.sciencedirect.com/science/article/pii/S0304399108002611>.

Answer to Comment 21: We appreciate the reviewer's insightful comments and valuable feedback aimed at improving the accuracy and reliability of our APT results. As you pointed out, compositional variations in APT can be influenced by field conditions, a well-documented issue since Miller's study on Fe-Si (1981). In our study, we prepared samples with nearly identical geometries and conducted analyses under similar voltage conditions, minimizing potential variations in field conditions. However, we acknowledge that we did not explicitly discuss this aspect in the original manuscript. To ensure that direct comparisons between different samples remain valid, we have examined the Si charge state ratios ($\text{Si}^{2+}/\text{Si}^+$) as an indicator of field conditions. The results of this comparison have been incorporated into the revised manuscript.

The manuscript has been revised as follows:

Please see Page #8.

“The SiON layer contained approximately 20 at.% N after a short PN time but more than 28 at.% after a longer PN time under similar charge state ratio (CSR) values (Supplementary Fig. 12). Because APT

field evaporation conditions can vary depending on the analysis environment, comparing data at similar CSR values—an indicator of comparable field conditions—is preferable for ensuring a reliable compositional analysis.^{18,64-66}

Please see Supplementary Fig. 12.

Supplementary Fig. 12 N concentration as a function of the CSR value (Si²⁺/Si⁺). At similar CSR values, the trends in the N concentration differ significantly depending on the PN & PNA process.

References have been added as follows:

18. Yeoh, W. K., Hung, S. W., Chen, S. C., Lin, Y. H. & Lee, J. J. Quantification of dopant species using atom probe tomography for semiconductor application. *Surf. Interface Anal.* **52**, 318–323 (2020). [10.1002/sia.6706](https://doi.org/10.1002/sia.6706).
64. Miller, M. K. & Smith, G. D. W. An atom probe study of the anomalous field evaporation of alloys containing silicon. *J. Vac. Sci. Technol.* **19**, 57–62 (1981). [10.1116/1.571017](https://doi.org/10.1116/1.571017).
65. Shariq, A., et al. Investigations of field-evaporated end forms in voltage- and laser-pulsed atom probe tomography. *Ultramicroscopy* **109**, 472–479 (2009). [10.1016/j.ultramic.2008.10.001](https://doi.org/10.1016/j.ultramic.2008.10.001)
66. Mancini, L., et al. Composition of wide bandgap semiconductor materials and nanostructures measured by atom probe tomography and its dependence on the surface electric field. *J. Phys. Chem. C.* **118**, 24136–24151 (2014). [10.1021/jp5071264](https://doi.org/10.1021/jp5071264).

Comment 22: ???

Answer to Comment 22: For clarity, we have changed "affords" to "results in" in the revised manuscript to improve readability and precision.

The manuscript has been revised as follows:

Please see Page #12.

“Hence, we confirmed that simultaneously adjusting the PN and PNA process resulted in a shallow N profile with a higher N concentration in the SiON layer.”

Comment 23: I think the authors should actually look into the literature. These effects coming from trajectory aberrations etc. are well known and well documented and these previous studies. should be cited... !

Answer to Comment 23: We apologize for the omission of relevant references in the manuscript. As you pointed out, local magnification effects and intermixing due to trajectory aberrations are well-documented in previous studies, and we should have included appropriate citations. Per your suggestion, we have added the relevant references in the revised manuscript to properly acknowledge prior research on this topic.

The manuscript has been revised as follows:

Please see Page #13.

“The layers along the vertical direction were widened and showed slight intermixing with neighboring layers owing to trajectory aberrations in the APT analysis, as well documented in previous studies;^{29,32,74-76}”

References have been added as follows:

29. Khan, M. A., Ringer, S. P. & Zheng, R. Atom probe tomography on semiconductor devices. *Adv. Mater. Interfaces* **3**, 1500713 (2016). [10.1002/admi.201500713](https://doi.org/10.1002/admi.201500713).
32. Chang, A. S. & Lauhon, L. J. Atom probe tomography of nanoscale architectures in functional materials for electronic and photonic applications. *Curr. Opin. Solid State Mater. Sci.* **22**, 171–187 (2018). [10.1016/j.cossms.2018.09.002](https://doi.org/10.1016/j.cossms.2018.09.002).
74. Larson, D. J., Gault, B., Geiser, B. P., De Geuser, F. & Vurpillot, F. Atom probe tomography spatial reconstruction: Status and directions. *Curr. Opin. Solid State Mater. Sci.* **17**, 236–247 (2013). [10.1016/j.cossms.2013.09.002](https://doi.org/10.1016/j.cossms.2013.09.002).
75. Beinke, D., Oberdorfer, C., & Schmitz, G. Towards an accurate volume reconstruction in atom probe tomography. *Ultramicroscopy* **165**, 34-41 (2016). [10.1016/j.ultramic.2016.03.008](https://doi.org/10.1016/j.ultramic.2016.03.008).
76. Hu, R., Xue, J., Wu, X., Zhang, Y., Zhu, H., & Sha, G. Atom probe tomography characterization of dopant distributions in Si FinFET: challenges and solutions. *Microsc. Microanal.* **26**, 36-45 (2020). [10.1017/S1431927619015137](https://doi.org/10.1017/S1431927619015137).

Comment 24: ?? Which one???

Answer to Comment 24: We appreciate the reviewer's comment and acknowledge the need for clarification. For instance, the previous study by Martin, A. J. et al. analyzed elemental distributions in a FinFET structure using APT (<https://www.sciencedirect.com/science/article/pii/S0304399117304084#fig0001>). In their study, TiN, rather than N itself, was visualized. N has been represented through the visualization of TiN distributions, making it difficult to distinguish N independently. As a side, to avoid redundancy, we have changed the sentence while retaining the key message in the revised manuscript.

The manuscript has been revised as follows:

Please see Page #13,

“Our APT analysis enabled the direct simultaneous observation of the 3D distributions of N, Si, and SiO_x, rather than representing N by visualizing the TiN distributions. The N₂⁺ peak was clearly distinguished from the Si⁺ peak extracted from a certain volume of the reconstructed APT ion map (Supplementary Fig. 17). The resulting N map, independent of the TiN map, provides valuable insights into the impact of the N distribution within actual 3D device structures, effectively visualizing its spatial distribution even in the presence of Si.”

Comment 25: I would expect to see composition profiles here please

Answer to Comment 25: We appreciate your suggestion and have addressed it by adding the composition profile from one fin to another fin in the 3D fin structure. Some degree of elemental intermixing due to trajectory aberrations is observed. Additionally, in the enlarged composition profile, the distribution and concentration of residues can also be identified. These updates have been incorporated into the revised manuscript.

The manuscript has been revised as follows:

Please see Fig. 7,

Fig. 7 APT analysis of the Si/SiO₂/TiN fin-structured device. **a** APT 3D reconstruction overlaid on the HAADF-STEM image. **b** Reconstructed APT 3D ion map after being cropped along the Y–Z plane to improve visualization. The Si at the SiO₂/TiN interface at an iso-concentration of 20 at.% is highlighted in red. **c** Ion maps of N (light green), O (blue), Si (orange), and Cl (cyan) in the fin-structured 3D device. **d** Composition profile from one fin to another fin (indicated by the bold arrow in **e**). Some degree of elemental intermixing occurred owing to the trajectory aberration. **e** Enlarged composition profile showing the distribution of residual Cl.

NCOMMS-24-82799A

RESPONSE TO REVIEWER'S COMMENTS

Reviewer #1 (Remarks to the Author):

General Comment: Authors have revised their manuscript satisfactory.

Answer to General Comment: We appreciate Reviewer #1's positive feedback and are pleased to hear that the revisions were satisfactory.

Reviewer #2 (Remarks to the Author):

General Comment: The authors have adequately addressed the reviewer's comments and concerns. I have a few minor suggestions for improving the manuscript.

Answer to General Comment: We thank Reviewer #2 for the thoughtful suggestions and have addressed the minor revisions to improve the clarity and presentation of the manuscript.

Comment 1: The authors mention using SIMS, STEM-EDS, and EELS techniques in the manuscript; however, detailed experimental procedures for these methods are missing from the "Methods" section. Including these details would improve clarity and reproducibility.

Answer to Comment 1: We sincerely appreciate your suggestion and fully agree that including methodological details is important for clarity and reproducibility. Accordingly, we have added specific descriptions of the experimental procedures for SIMS, STEM-EDS, and EELS analyses in the revised "Methods" section. These include details on instrumentation, operating conditions, and analysis parameters. We hope this addition helps readers better understand and replicate the complementary characterization methods used in our study.

The manuscript has been revised as follows:

Please see Page #15,

"Scanning transmission electron microscopy analysis

Cross-sectional specimens for STEM observations were fabricated using a focused ion beam (FIB Helios5 HX, Thermo Fisher Scientific) and milled at 5 and 2 kV at currents of 16 and 3 pA, respectively, to prevent damage from the Ga-ion beam. The cross-sectional microstructures of the thin SiO₂/SiON/TiN and ultra-thin SiO₂/SiON/poly-Si structures were observed using spherical aberration (Cs)-corrected STEM (FEI-Titan Cubed) at 300 kV. STEM-EDS mapping was conducted using an electron beam current of 70 pA, with a total acquisition time of 600 s at a magnification of 1.3 M. STEM-EELS analysis was carried out using a Gatan Imaging Filter spectrometer in STEM mode with a convergence angle of 26 mrad and a collection angle of 43 mrad. Elemental mapping was performed

in spectrum imaging mode with an energy dispersion of 0.05eV/channel, an acquisition time per pixel of 0.2 to 1 s, and a pixel size of 0.23 nm at a camera length of 29.5 cm. An electron beam current of 50 pA was used to minimize the electron beam-induced damage. The absence of the electron beam-induced damage was verified before and after EELS acquisition.”

“Secondary-ion mass spectrometry analysis

SIMS analysis was performed with an IONTOF M6 instrument (ION-TOF GmbH, Muenster, Germany) operated in dual-beam mode. A 30 keV Bi⁺ primary ion beam with a current of 1.0 pA was employed as the analysis beam. A 1keV Cs⁺ sputter beam with a current of 20 nA was used for depth profiling in negative ion mode, creating a 300 × 300 μm² sputter crater. Data were collected at 128 × 128 pixels over a 100 × 100 μm². A flood gun for charge compensation was not applied.”

Comment 2: Figure 4: The micron scale marker in the STEM image is not legible and should be improved for better visualization.

Answer to Comment 2: We sincerely appreciate your thoughtful suggestion and fully agree. The scale bars in Fig. 4 have been revised to improve legibility and ensure better visualization.

The manuscript has been revised as follows:

Please see Fig. 4,

(Scale bars: 5 nm in a, b).

Comment 3: Figure 6: The STEM-HAADF image lacks a micron scale marker, which should be added.
Answer to Comment 3: We apologize for the omission and have added the missing scale bars in Fig. 2.

The manuscript has been revised as follows:

Please see Fig. 6.

(Scale bars: 2 nm in **a**, **b**).

Comment 4: Figure 7: In the APT analysis of the Si/SiO₂/TiN fin-structured device, the color coding for elements (Si, O, N) appears inconsistent with earlier figures. This inconsistency may lead to confusion; a uniform color scheme across all figures is recommended for clarity.

Answer to Comment 4: We sincerely thank the reviewer for pointing this out. Although we originally used slightly different color schemes in Fig. 7 to enhance visualization, we agree that maintaining consistency across figures is important for clarity. We have now revised the color coding in Fig. 7 to match that of the earlier figures. If there still appears to be a slight visual difference, it may be due to the rendering style—whether atoms are displayed as points or spheres—which was chosen to optimize visualization. We hope this is understandable and appreciate your thoughtful feedback. Please note that Supplementary Fig. 1 has been renamed as Supplementary Table 1, which has resulted in a shift in the numbering of the subsequent Supplementary Figures. We kindly ask for your understanding regarding this adjustment.

The manuscript has been revised as follows:

Please see Fig. 7.

(Scale bars: 20 nm in **a-c**).

Please see Supplementary Fig. 13.

(Scale bar: 5 nm).

Comment 5: Supplementary Figures 10 and 11(c, d): These figures lack clarity, and the presented information is difficult to discern. Enhancing the resolution, axis labels, and contrast would significantly improve readability and the overall quality of the presentation.

Answer to Comment 5: Thank you for your valuable feedback. We fully agree with your suggestion. To improve clarity and readability, we have enhanced the resolution of Supplementary Figures 10 and 11(c, d), adjusted the axis labels for better visibility, and improved the contrast of the plots. Please note that Supplementary Fig. 1 has been renamed as Supplementary Table 1, which has resulted in a shift in the numbering of the subsequent Supplementary Figures. We kindly ask for your understanding regarding this adjustment.

The manuscript has been revised as follows:

Please see Supplementary Fig. 9.

Please see Supplementary Fig. 10.

(Scale bars: 5 nm in **b-d**).

Reviewer #3 (Remarks to the Author):

General Comment: I am satisfied with the reply to my comments and the associated modifications to the manuscript. IMHO, the paper can now be published.

Answer to General Comment: We appreciate Reviewer #3's helpful comments and are glad that the manuscript is now considered suitable for publication.